

# State Updating and Calibration Period Selection to Improve Dynamic Monthly Streamflow Forecasts for a Wetland Management Application

Matthew S. Gibbs[1,2], David McInerney[1], Greer Humphrey[1], Mark A. Thyer[1], Holger R. Maier[1], Graeme C. Dandy[1], Dmitri Kavetski[1]

[1]School of Civil, Environmental and Mining Engineering, The University of Adelaide, North Terrace, Adelaide, South Australia, 5005, Australia
[2]Department of Environment, Water and Natural Resources, Government of South Australia, PO Box 1047, Adelaide, 5000.

*Correspondence to*: Matthew Gibbs (matthew.gibbs@adelaide.edu.au)

**Abstract.** Sub-seasonal streamflow forecasts provide useful information for a range of water resource management and planning applications. This work has focused on improving forecasts for one such application: the management of water available in an open channel drainage network to maximise environmental and social outcomes in a region in southern Australia. Conceptual rainfall-runoff models with a postprocessor error model for uncertainty analysis were applied to provide forecasts of monthly streamflow. Two aspects were considered to improve the accuracy of the forecasts: 1) state updating to force the models to match observations from the start of the forecast period, and 2) selection of a calibration period representative of the forecast period. Five metrics were used to assess forecast performance, representing the reliability, precision, bias and skill of the forecasts produced, using both observed and forecast climate data. The results indicate that assimilating observed streamflow data into the model, by updating the storage level at the start of a forecast period, improved the performance of the forecasts across the metrics when compared to an approach that "warmed up" the storage levels using historical climate data. The shorter calibration period improved the performance of the forecasts, particularly for a catchment that was expected to have experienced a change in the rainfall-runoff relationship in the past. The results highlight the importance of identifying a calibration record representative of the expected forecast conditions, and if this step is ignored degradation of predictive performance can result.



# 1 Introduction

Predictions of streamflow a month or a season ahead are essential information required by water resource managers for subsequent planning (Wang et al., 2011). This is particularly true in unregulated catchments with no capacity for storage and a highly variable flow regime that can be difficult to predict from historical data. This is the case for the Drain M catchment in southern Australia, where streamflow can be diverted to support a range of environmental and social outcomes, and improved information on future water availability can assist in maximising the benefits realised from the water available. A number of approaches have been developed to provide streamflow predictions with lead times from a month to a season ahead. These include hydrological modelling approaches (Demargne et al., 2014; Wood and Schaake, 2008), statistical approaches (Bennett et al., 2014; Robertson and Wang, 2013), or a combination of the two (Robertson et al., 2013).

In this work, a hydrological modelling based approach has been adopted to provide streamflow forecasts for an environmental management application, as this approach can often capture catchment dynamics that the simple indices used in statistical models cannot (Robertson et al., 2013). In forecast mode, a hydrologic model that has been calibrated with historical data is run forward in time, with input data representing forecast climate forcings, to predict the streamflow in an upcoming period. The following three major factors control forecasting performance (Luo et al., 2012): (1) the ability of the hydrologic model to predict streamflow with actual forcings; (2) the accuracy of the initial conditions adopted (e.g., soil moisture stores); and (3) the accuracy of the forecasts of the climate inputs. The focus of this paper is on the first two factors related to the practical application of climate forecasts, in the context of a user need for seasonal forecasts to assist with answering environmental management questions. The accuracy of the climate forecast is an important component of streamflow forecasts, and the effect of climate forecasts on streamflow forecasts is also considered in this work, but not approaches to improve climate forecasts.

## 1.1 Ability of the hydrologic model to predict streamflow

Conceptual rainfall-runoff (CRR) models are widely used to simulate streamflow, due to their simplicity and accuracy (Tuteja et al., 2011). As these models are conceptual, they require calibration to observed streamflow (if available). It is generally considered that longer calibration periods produce more robust parameter estimates (Brigode et al., 2013). However, the use of long calibration periods assumes time-invariant catchment characteristics and processes, and that the parameter values derived from the calibration period are representative for the prediction period (Vaze et al., 2010).

The fitting of constant parameter values based on a given set of observed data can result in decreased model performance over time if the conditions that the model is exposed to in the forecast period are different from those used during model calibration (Coron et al., 2012). For the purposes of this work, catchments where physical changes are expected to have occurred, and there is evidence to reject the hypothesis of stationarity of the physical system, have been termed "non-stationary". Examples of non-stationary changes in catchment conditions that could be expected to change the rainfall-runoff relationship include changes in land use or land-cover (e.g., deforestation, urbanization), land drainage, interception (e.g. dams or diversions),





groundwater depletion or responses to changes in climate (Milly et al., 2015). This definition of non-stationarity, or catchment non-stationarity, is used in this work, in contrast to a broader definition of "hydrological model non-stationarity" which is used when hydrological model parameters vary in time for any reason (e.g. systematic data errors, poor calibration procedures, model structural deficiencies), and are therefore dependent on the period of record used for their estimation (for example,

Westra et al., 2014).

Degradation in performance due to catchment non-stationarity can have a significant influence on the streamflow simulated by the model, and hence on the decisions informed by subsequent forecasts. In order to address this concern, a number of studies have proposed calibrating model parameters to subsets of the available data, by finding analogous periods in the historical record to compensate for deficiencies in the model structure or input data (Brigode et al., 2013; de Vos et al., 2010;

Luo et al., 2012; Vaze et al., 2010; Wu et al., 2013; Zhang et al., 2011). The impact of calibration period on model results, including predictive uncertainty, is one aspect considered in this study.

For the purposes of predicting streamflow, it is desirable for the hydrological model to not only be able to suitably predict streamflow with actual forcings, but also quantify the uncertainty associated with the prediction. Many approaches are available to quantify the uncertainty in model predictions, from identifying a range of model parameters that represent the

behaviour of the catchment using approaches such as generalised likelihood uncertainty estimation (GLUE) (Beven et al., 2008), to disaggregation approaches that attempt to characterise each individual source of error explicitly (e.g. Kavetski et al., 2003; Vrugt et al., 2005). In this work, predictive uncertainty has been estimated using a postprocessor error model. This error model provides a statistical description of the differences between the model predictions and observed data, without trying to identify the contributing sources (Evin et al., 2014). This approach is generally less data intensive and less complex than

disaggregation approaches, which attempt to characterise each individual source of error. A post-processor approach to error modelling can also lead to more robust estimates of the model uncertainty compared to jointly assessing all parameters (i.e. estimating CRR model and error model parameters concurrently) (Evin et al., 2014).

Adequate representations of errors is necessary to obtain reliable and precise hydrological predictions. The term "reliable" is used to define predictions that are statistically consistent with the distribution of the observations, and "precise" defined as

predictions with a tight range of uncertainty. If poor assumptions are made about the properties of errors, unreliable or highly uncertain predictions can be obtained (Thyer et al., 2009). For example, errors in hydrological applications are generally heteroscedastic (i.e. the variance in the model errors is not constant, and increases with the magnitude of the flow). (McInerney et al., 2017) provided guidance on approaches most suitable for accounting for heteroscedasticity for different catchment types.

## 1.2    Accuracy of the initial conditions

Much of the skill in seasonal streamflow forecasts over periods following rainy seasons is attributed to accurately representing initial catchment conditions (Koster et al., 2010; Pagano et al., 2004; Wang et al., 2009), whereas forecast skill over periods



following dry seasons is attributed to both initial catchment conditions and meteorological inputs (Maurer and Lettenmaier, 2003; Wood and Lettenmaier, 2008). The impact of initial catchment condition is particularly pronounced when forecasting over short lead times, up to one month (Li et al., 2009; Wang et al., 2011).

In CRR models, catchment conditions are represented by (usually multiple) model storages, which are referred to as state variables. The values of these storages at the start of forecast period are typically determined using a "warmup" period, which allows the internal model states to reach reasonable values (e.g. Wang et al., 2011). Given the expected influence of the initial conditions on the simulated streamflow, model storage updating using observed data can be adopted to determine the initial state of the model storages (Wöhling et al., 2006). However, updating model storages of conceptual rainfall-runoff models is not straightforward, as the model states are at best approximate representations of the real world as seen by the model (Berthet et al., 2009). A number of potential observed data sources could be used to update model storages. Li et al. (2015) suggested that although advanced remote sensing techniques provide an opportunity to improve hydrologic simulation, gauged discharge data assimilation is a more effective way to improve short-term forecasts and is still preferred for operational streamflow forecasting purposes.

The impact of assimilating observed data into CRR model state variables at the start of a forecast on probabilistic seasonal forecasts has had limited evaluation. It could be expected that by assimilating this extra information into the model state variable(s) the precision of the forecast, and potentially its reliability, could be increased.

### 1.3    Study Aims and Objective

The objective of this work is to determine the degree to which post-processing and calibration periods can enhance monthly streamflow forecasting skill for the management of water availability for an environmental flow management application. The specific aims to achieve this objective are:

- Evaluate a data assimilation approach for CRR model state updating to enhance forecasting skill,
- Demonstrate that calibration period can affect forecast skill, and can be considered as a mechanism to account for non-stationarity in a catchment.
- Demonstrate a postprocessor error model for uncertainty analysis and quantify forecasting skill (reliability, precision, bias).

In the following section, a user need for seasonal forecasts to manage a drainage network for environmental and social outcomes in Southern Australia is outlined, including the catchments of interest and data available. This is followed by the methodology adopted to demonstrate the aims outlined above.



## 2    Case Study for Streamflow Forecasts

The location considered in this study is a component of an extensive drainage network (exceeding 2500 km of open channels) in the South East of South Australia. A large proportion of the region's runoff is generated in the higher rainfall areas of the Lower South East. Historically, this runoff flowed in a northerly direction, along the watercourses adjacent to ranges, parallel to the coastline. Over the past 150 years, these flow paths were diverted through a series of cross-country drains, constructed to provide flood relief and improve the agricultural productivity of the region by draining water in a south-westerly direction, creating outlets to the ocean. Drain M, which conveys water from Bool Lagoon to the ocean near Beachport (from station A2390541 to A2390512, Figure 1), is the largest of the cross-country drains. This drain collects flow from several drainage systems located to the east and south of the drain, as outlined in Figure 1.

Recently, the ability to divert flow from this drain to the Upper South East and toward Ramsar listed wetlands (the Coorong) has been established. This diversion point is the Callendale regulator, located just upstream of gauge A21390514 which can divert flow to the north and restore a more natural flow path (as indicated by the drainage line in Figure 1). However, the termination of the Drain M system also contains Lake George, a water body of high importance. Lake George has a high social value, including a fishery and public access, and can suffer from reduced water quality (hyper-salinity) without sufficient volumes from the drainage network.

Decisions must be made throughout the year (mainly in the high flow season from late winter and throughout spring) regarding diversions from Drain M; if water should continue to flow to Lake George and the ocean, or if it can be diverted to the north, and be used to support inundation at numerous wetlands. Such decisions aim to meet the following competing objectives:

- Ensure the water requirements of Lake George are met. Freshwater is necessary to maintain the estuarine ecology of the lake, to support its significance as a biological resource, and as a resource for recreational fishing.
- Maintain some flow to the ocean to restrict sediment from entering the lake and to maintain connectivity to the sea to allow for fish movement and aide fish recruitment.
- Minimise possible impacts on sea grasses, caused by fresh water flows, often with high nutrients, out to sea.
- Maximise water diverted to the north, to benefit the wetlands in the Upper South East, subject to the downstream objectives at Lake George.

Runoff volumes from Drain M are highly variable, ranging from close to zero to more than is required to support Lake George, with the historical volumes varying over 3-4 orders of magnitude for a given month. This variability makes it difficult to maximise the use of water, as the seasonal pattern described by the historical record alone provides little guidance. Bool and Hacks Lagoons (between stations A2390519 and A2390541 in Figure 1), act as water storages in this region as well as being significant Ramsar listed wetlands. With reduced inflows and the need to meet the water requirements of Bool and Hacks Lagoons, only minor releases from the lagoons at station A2390541 have occurred in the past 15 years. With the many





competing demands on the water resources available in this system, it is desirable to forecast future flows at key locations along Drain M to maximise the environmental and social outcomes achieved from the water available.

To assess potential benefits and impacts of different diversion scenarios on the wetlands in the Drain M system, a water balance model has been developed in eWater Source (Welsh et al., 2013). The model also simulates salinity in Lake George based on

a constant inflow of salinity from Drain M, and modelling the subsequent flushing and evapoconcentration of the lake. This model provides the functionality to take forecast streamflow volumes in the drainage network and allow different management scenarios to be considered for the main operational locations in the system: Bool Lagoon and Callendale regulator. Given a probabilistic streamflow forecast, the impact of different diversions and releases on the forecast range of water level in Bool Lagoon, water level and salinity in Lake George and volume diverted along the drain can be estimated. This capability is

intended to provide greater information to assist the decision-making process, extending the probability of a certain volume of water occurring in the network to enable the influence of operational decisions on the key variables (volume diverted, water levels and Lake George salinity) in the system to be considered. The model has been calibrated to provide a suitable representation of historical water levels, flows and salinities.

To use this model for to inform operations, forecast climate and streamflow are required model inputs. The focus of this work

is to derive suitable streamflow forecasts for this application. A forecast period with a leadtime of one month has been considered as most appropriate for this application, as 1) the main quantities of interest in this application are volume and the water balance rather than the size or timing of peak flows, 2) one month provides sufficient time to undertake any changes in management, and 3) reasonable forecast skill is expected to be possible compared to longer forecast horizons.

### 2.1    Climate and Climate Data

The mean annual rainfall for the region is in the range 600 - 675 mm and the mean annual FAO56 potential evapotranspiration (PET) (Allen et al., 1998) is approximately 1000 mm. The highest rainfalls are experienced in the winter months, with rainfall exceeding evapotranspiration in the months May – September. The SILO Patched Point Dataset (Jeffrey et al., 2001) was used for the rainfall and FAO56 evapotranspiration data adopted, with the climate stations used shown in Figure 1. A Thiessen polygon approach was used to combine stations and produce one time series each of rainfall and evapotranspiration for each

catchment. This weighting approach has been considered appropriate for the region, due to the flat terrain being unlikely to lead to significant topographic effects on the spatial distribution of rainfall.

Rainfall forecasts from the Australian Bureau of Meteorology's (BoM) seasonal forecast system, POAMA-2, were used to represent the effect of climate during the forecast period. POAMA is a dynamical climate forecasting system designed to produce multi-week to seasonal forecasts of climate for Australia based on a coupled ocean/atmosphere model and

ocean/atmosphere/land observation assimilation systems. POAMA-2 consists of both a seasonal and a monthly/multi-week forecast system using version 2.4 of the model. Both systems are multi-model ensembles consisting of three different model configurations. Furthermore, each configuration is used to generate an ensemble of 10 forecasts by perturbing initial





conditions, resulting in a 30-member ensemble. POAMA-2 produces large (~250 km) spatial scale predictions over Australia, which do not capture the large degree of spatial variability in rainfall. For application in this study, the large scale POAMA-2 rainfall hindcasts (i.e. forecasts developed using the modelling system for the historical period) were downscaled to the single rainfall gauge scale. The statistical downscaling method detailed in Shao and Li (2013) was used, resulting in rainfall

hindcasts at each of the rain gauges shown in Figure 1. As an example Figure 2 shows that the mean rainfall hindcasts at station 26000 provide a reasonable estimate of the observed rainfalls at this station, while the upper and lower hindcast bounds capture the majority of observed rainfalls. Further details of the downscaling approach are provided in Humphrey et al. (2016).

## 2.2    Streamflow and Streamflow Data

The streamflow in the region is seasonal to ephemeral, with flow ceasing over the summer and autumn months and runoff

coefficients are low, with annual runoff in the range of 2-10% of annual rainfall (Gibbs et al., 2012). The predominant land use in the region is dry land pasture with some flood irrigation as well as plantation forestry; there is no major urbanization in the catchments. The topology of the region is very flat, with mainstream slopes in the order of 0.005. The hydrogeology of the catchment includes shallow aquifers with major karstification of limestone, which may be suggestive of groundwater exchange across the boundaries of the catchments.

The details of the three catchments contributing to Drain M are as follows. Mosquito creek flows into Bool Lagoon (catchment area of 1002 km$^2$), with the inflows gauged at station A2390519 (referred to as C1, as shown in Figure 1). The Southern Bakers Range catchment flows into Drain M between Bool Lagoon and the diversion point at Callendale (2200 km$^2$ – C3). The flow from this catchment has been determined from that gauged at Callendale (A2390514) minus that released from Bool Lagoon (A2390541). Travel times along this section of drain are less than one day, and for the purposes of this study the daily flows

have been subtracted to determine the flow from the Southern Bakers Range catchment. It should also be noted that releases from Bool Lagoon have occurred infrequently in the recent past, with only four short release events since 1997 (Gibbs et al., 2014). Finally, the Drain M local catchment contributes flow downstream of the Callendale diversion point, flowing into Lake George (383 km$^2$). This is gauged at station A2390512 (C2). As with the Southern Bakers Range catchment, the flow contributing to the drain from this local catchment has been derived as the difference between the flow at stations A2390514

and A2390512, as travel times are again less than one day.

Daily streamflow data are available from the South Australian Department of Environment, Water and Natural Resources Surface Water Archive (https://www.waterconnect.sa.gov.au/Systems/swd), with the stations used shown in Figure 1. Three of the stations have data available from the early 1970s, with the exception being the station at the outlet of Bool Lagoon (site A2390541), where data were available from 1985.

Streamflow measurements in the three catchments were obtained from fixed concrete weirs. The stations at A2390512 and A2390519 have v-notch cross-sections to improve sensitivity at low flows, whereas stations A2390514 and A2390541 have



flat cross sections and therefore could be expected to be less sensitive at low flows. All stations are free flowing downstream and are constructed drains upstream with a constant slope and cross section, which would be expected to provide a reliable relationship between stage and discharge. Gauging records for the four stations are extensive, with between 78 and 166 flow gaugings at each station, with a 90th percentile deviation of 10.5% for stations A2390512 and A2390519, and 16.3% at station

A2390514. Given the regular cross sections and high number of gaugings available to develop stage-discharge relationships, output data uncertainty is expected to be low for the stations considered. This identification of high quality data was considered an important process, as biases and systematic changes in the measurement of hydrological data can significantly affect model calibration and can lead to non-stationarity in the estimated model parameters (Westra et al., 2014).

In the region where the case study catchments are located, plantation forestry expanded substantially in the late 1990s. Changes

in the relationship between rainfall and runoff also occurred during this period, as seen as the reduced slope in the double-mass analysis (Searcy et al., 1960; Yihdego and Webb, 2013), i.e. the plot of cumulative runoff against cumulative rainfall in Figure 3. These changes have implications for the period most relevant for forecasting future flows in the catchments, as data from the 1970s may not be representative of what could be expected in future years. It is also evident from Figure 3 that, despite the contributing area, very little runoff is generated from C3 due to a number of factors, including the very flat terrain

and substantial depression storage, substantial vegetation cover (both plantation and natural) and irrigation use from the shallow underlying aquifer. Given the limited volumes generated within this catchment, it has been excluded from further analysis. From a practical perspective, in years where there is substantial yield from this catchment, it is assumed that there will already be surplus flow from the upstream catchments. Based on the case study outlined, the two main challenges considered in this paper are: (i) estimation of predictive uncertainty to inform management, and (ii) determination of

approaches to adequately represent the non-stationarity in the catchments.

## 3    Methodology

### 3.1    CRR Model and Parameter Calibration

The GR4J model (Perrin et al., 2003) was selected as the CRR model for this study, as it is a parsimonious model that explicitly accounts for non-conservative (or 'leaky') catchments, and has demonstrated good performance for Australian conditions

(Coron et al., 2012; Guo et al., 2017). The ability to represent non-conservative catchments is important in the study area (see section 2.2). The standard form of the GR4J model has four calibration parameters, the maximum capacity of a production (soil) store, X1, a catchment water exchange coefficient, X2, the maximum capacity of a routing store, X3, and a time base for a unit hydrograph, X4. Further details of the model structure and parameters can be found in Perrin et al. (2003). In addition to the four standard parameters, the split between the routing store and the direct runoff has also been considered a calibration

parameter in this study. The resulting additional parameter is termed *split*; the standard GR4J formulation adopts a default value of *split*=0.9.



## 3.2 Parameter estimation

The GR4J parameters were inferred using Bayes equation. The posterior probability density of the parameters given daily observed streamflow data $\widetilde{\boldsymbol{q}}$ and climate data $\mathbf{X}$, $p(\boldsymbol{\theta}|\widetilde{\boldsymbol{q}},\boldsymbol{X})$, given by:

$$p(\boldsymbol{\theta}|\widetilde{\boldsymbol{q}},\boldsymbol{X}) \propto p(\widetilde{\boldsymbol{q}}|\boldsymbol{\theta},\boldsymbol{X})\,p(\boldsymbol{\theta}) \qquad (1)$$

where $p(\boldsymbol{\theta})$ is the prior distribution and $p(\widetilde{\boldsymbol{q}}|\boldsymbol{\theta},\boldsymbol{X})$ is the likelihood function.

A standard least squares likelihood function was adopted (see, for example, Thyer et al., 2009). The function was calculated using the daily time series, and is derived from an error model assuming independent, homoscedastic residuals. While the assumptions required by this function (i.e. constant variance, independent) are somewhat unrealistic for hydrological applications, statistics based on sum of squared errors (Nash Sutcliffe Efficiency, RMSE) are often used for model calibration

and assessment, and hence this function has been selected. This function has also been adopted given that the focus of this study is on the estimation of monthly runoff volumes, as this function provides a focus on the highest flows in the time series, where the majority of the runoff volume occurs (Wright et al., 2015).

Uniform prior distributions were adopted for all parameters, with the assumed bounds given in Table 1. A number of trial runs were used to ensure the bounds of the prior distribution were suitable and well-removed from the high-density areas of

the posterior distribution.

The posterior distribution in equation (1) was sampled using the DiffeRential Evolution Adaptive Metropolis (DREAM) algorithm (Vrugt et al., 2009). DREAM provides a sample of parameter sets that represents the posterior parameter distribution, and the trends in these distributions for different time periods are considered in the results. For the purposes of residual error modelling, only the parameter set resulting in the maximum posterior was used. The Hydromad R package (Andrews et al.,

2011) implementation of the DREAM algorithm and GR4J model have been adopted. A maximum of 25,000 evaluations were used, including a burn-in period where the initial samples were discarded before the Markov Chain was assumed to have stabilised. The number of parallel chains was set to the number of parameters, i.e. five parallel chains were adopted (Vrugt et al., 2009).

## 3.3 Calibration Periods to account for Non-stationarity

A rolling calibration approach has been used to minimize the impact of external influences on the inferred model parameters, similar to that of Luo et al. (2012) and Wagener et al. (2003). External influences include model structural limitations, physical changes in the catchments influencing the rainfall-runoff relationships (land use change, change in groundwater level), or over-representation of particular hydroclimatic conditions. In the rolling calibration approach, the model parameters were recalibrated each year based on the data from the preceding period. The parameter values identified were used to simulate the

following one year of data (following that used in the calibration process), before recalibrating the model to include these



"new" data, while discarding the earliest year of data, to maintain the same period length for calibration. The recalibration methodology adopted allows for the change in parameter distributions over time to be included.

10-year and 20-year calibration periods have been considered, to assess the trade-off between a longer calibration period to reduce the parameter uncertainty, and a shorter calibration period representing the most recent hydrological dynamics observed in the catchment. As an example, a 10 year calibration period was from 1/5/1995-30/4/2005, after a one year warmup period. The prediction period was then considered to be the monthly forecasts over the following year (1/5/2005-30/4/2006). The process was then repeated each year, where in this example the next calibration period becomes 1/5/1996-30/4/2006 and the prediction period 1/5/2006-30/4/2007. The start month of May corresponds to the start of the flow season in autumn. Through this rolling approach, the model predictions assessed were independent of the data used for calibration in all the results presented. For the case of using forecast rainfall, the model was run with observed rainfall up to the forecast month, before using forecast rainfall only for the forecast month. This was repeated for each month in the prediction period.

### 3.4    State Updating in GR4J

An approach to directly correct the states of a rainfall-runoff model with observed data has been adopted in this work, similar to that of Demirel et al. (2013). The model was run for a one year warmup period, and the simulated level in the production store at the end of this period was maintained. As model states generally do not reflect reality directly (Berthet et al., 2009), the production store level has been maintained "as seen" by the model.

In order to force the model to simulate the flow observed at the start of the month, the necessary routing store level to produce the observed flow after accounting for the modelled direct flow was calculated (Demirel et al., 2013). In GR4J, total simulated streamflow on a given day $q_t^\theta$ is calculated as the sum of the flow direct from the production store (after applying a unit hydrograph), $q_{t,d}^\theta$, and the flow from the routing store, $q_{t,r}^\theta$, i.e.:

$$q_t^\theta = q_{t,d}^\theta + q_{t,r}^\theta \qquad (2)$$

The necessary flow from the routing store for $q_t^\theta$ to equal the observed flow, $\widetilde{q}_t$, defined as $q_{t,r}^{SU}$, is then calculated as:

$$q_{t,r}^{SU} = \min(\widetilde{q}_t - q_{t,d}^\theta, 0) \qquad (3)$$

The routing store level, R, can then be obtained by setting $q_{t,r}^\theta = q_{t,r}^{SU}$ and inverting the equation used by the GR4J model to calculate the outflow from this storage:

$$q_{t,r}^\theta = R\left(1 - \left(1 + \left(\frac{R}{X3}\right)^4\right)^{-1/4}\right) \qquad (4)$$

where X3 is an estimated runoff model parameter.



More complex approaches for data assimilation and state updating are available, for example, the ensemble Kalman filter and particle filter approaches are commonly used (He et al., 2012; Lei et al., 2014; Li et al., 2013; Plaza Guingla et al., 2013; Spaaks and Bouten, 2013; Xie and Zhang, 2013). However, particularly when used to update both model state variables and model parameters, these approaches impose a substantial computational burden for obtaining accurate results, and accuracy

can decrease if incorrect correlations between states and parameters are identified, due to biased model error quantification and a large degree of freedom for high-dimensional vectors of the augmented state (Xie and Zhang, 2013).

### 3.5 Estimation of Predictive Uncertainty

A post-processor error model was used to estimate predictive uncertainty in forecasts. After the hydrological model has been calibrated, a statistical model was fitted to the residuals given by the difference between monthly observations ($\widetilde{\boldsymbol{Q}}$) and

predictions from the hydrological model ($\boldsymbol{Q}^{\theta}$) using the maximum posterior parameter estimates. In this case, the streamflow at the monthly time scale was considered, i.e., the sum of the streamflow at the daily time scale each month, $\widetilde{Q_t} = \sum_{i=1}^{n} \widetilde{q_t}$, for the n days in month t. Depending on the case, the hydrological model predictions were either those forced by observed rainfall, or for the case when the model was forced by the ensemble of forecast rainfall, the hydrological model predictions were the median across the ensemble of the hydrological model predictions each day, and then aggregated to the monthly time step.

Henceforth, the resulting hydrological model prediction monthly time series is referred to as $\boldsymbol{Q}^{\theta}$.

Heteroscedasticity (i.e. larger residuals for larger flows) and skewness in these residuals are captured by using the Box Cox transformation:

$$Z_{\mathrm{BC}}(Q;\lambda,A) = \begin{cases} \dfrac{(Q+A)^{\lambda}-1}{\lambda} & \text{if } \lambda \neq 0 \\ \log(Q+A) & \text{otherwise} \end{cases} \tag{5}$$

where $\lambda$ is a transformation parameter, and $A$ is an offset parameter (often important when transforming low flows). Based

on the findings of McInerney et al. (2017), $\lambda = 0.5$ was selected, which was shown to produce good predictive performance in ephemeral catchments, especially in terms of improving precision and reducing bias. A value of $A = 1 \times 10^{-5}$ mm/month was selected.

The Box Cox transformation is applied to observed and predicted flows, and normalized residuals are calculated as:

$$\eta_t = \mathrm{Z}(\widetilde{Q_t}) - \mathrm{Z}(Q_t^{\theta}) \tag{6}$$

These normalized residuals $\eta_t$ are assumed to be Gaussian with mean $\mu_\eta$ and variance $\sigma_\eta^2$, i.e. $\eta_t \sim N(\mu_\eta, \sigma_\eta^2)$.





The parameters $\mu_\eta$ and $\sigma_\eta$ are then estimated using the method of moments, i.e. as the sample mean and sample standard deviation of $\eta$ computed with the maximum posterior parameter set.

Once the residual error model has been calibrated, replicates from the predictive distribution, $\mathbf{Q}^{(r)}$ for $r = 1..N_r$, are generated for the independent evaluation period as follows:

1.    Sample the normalized residual at time step $t$, $\eta_t^{(r)} \leftarrow N(\mu_\eta, \sigma_\eta^2)$.            (7)

       2.    Rearrange Equation 6 to yield:

$$Q_t^{(r)} = Z^{-1}\left(Z(Q_t^\theta) + \eta_t^{(r)}\right)$$

(8)

       3.    Truncate negative values to zero.

The replicates represent the predictive distribution (PD) of the forecasts.

**3.6**      **Performance Metrics**

Five metrics were used to evaluate different aspects of predictive performance. These include metrics for reliability, precision, volumetric bias, the cumulative ranked probability score (CRPS) and Nash Sutcliffe Efficiency (NSE). Depending on the data available, the metrics are calculated over different periods. When observed climate data was used to drive the CRR models, the periods commences 21 years after the start of the streamflow record (after a 1 year warmup and 20 year calibration period),

that is 1/5/1992 for C1 and 1/5/1994 for C2. In all cases, the period ends on 30/4/2010. The same period is used within each case considered, e.g. different calibration lengths, with/without state updating, and as such are comparable within these cases for a given catchment. For display purposes enabling a straight forward comparison across all the cases considered, the value for each metric has been normalised by linearly scaling the worst value to a value of 0.05 and the best value to 0.95.

**Reliability** refers to whether observations over a series of time steps can be considered to be statistically consistent with being

samples from the predictive distribution. Reliability has been assessed using predictive quantile-quantile (PQQ) plots and the plot has been summarized using the reliability metric of Renard et al. (2010) which is based on the area between the PQQ plot and the 1:1 line. A value of 0 represents perfect reliability, while a value of 1 represents the worst case of all observations lying either above or below the PD.

**Precision** refers to the width of the predictive distribution, and is otherwise known as resolution or sharpness. We quantify

precision using the following metric from McInerney, et al (2017):



$$\text{Precision} = \frac{1}{N_t}\sum_{t=1}^{N_t}\text{sdev}\,\mathbf{Q}_t \;\bigg/\; \frac{1}{N_t}\sum_{t=1}^{N_t}\tilde{Q}_t \tag{9}$$

**Volumetric bias** measures the overall water balance error of the predictions relative to the observations. It is calculated as:

$$\text{VolBias} = \left| \frac{\sum_{t=1}^{N}\text{mean}\,\boldsymbol{Q}_t^\theta - \sum_{t=1}^{N}\widetilde{Q_t}}{\sum_{t=1}^{N}\widetilde{Q_t}} \right| \tag{10}$$

**CRPS** is a widely used metric that summarises and evaluates multiple aspects of predictive performance (including reliability,
uncertainty and sharpness) in a single measure (Hersbach, 2000). The CRPS is calculated by comparing the cumulative
distribution of the forecast with the cumulative distribution of the observation at each time step. For each time step, the CRPS
is calculated as:

$$\text{CRPS} = \int_{-\infty}^{\infty}\big[F_f(Q_t) - F_o(Q_t)\big]^2\,dQ \tag{11}$$

where $F_f$ and $F_o$ are the cumulative distributions of the streamflow forecast and observation, respectively, at time step $t$. The
average value of the CRPS is then calculated over all time steps $t$. Note that the cumulative distribution of the observations is
a step function. A CRPS of 0 corresponds to a perfect forecast, while larger CRPS values correspond to worse performance.

To normalize CRPS metric values across catchments, the CRPS metric for the forecast (CRPS$_F$) is expressed as a skill score
with respect to the CRPS metric of a reference distribution for that catchment (CRPS$_R$)

$$\text{CRPS}_{SS} = \frac{\text{CRPS}_R - \text{CRPS}_F}{\text{CRPS}_R}$$

A CRPS$_{SS}$ of 0 corresponds to the forecast having similar performance as the reference distribution, while a CRPS$_{SS}$ of 1
corresponds to a perfect forecast.

The reference distribution for each month is calculated as the empirical distribution of all observed data in that month. Note
that since all of the observed data is used to estimate the reference distribution (including data in the forecast verification
period), this distribution has an unfair advantage over the models which use only previously observed data. This means that
there may be cases where the CRPS$_{SS}$ is less than zero, corresponding to the reference distribution having better performance
than the forecast.

**NSE** is a commonly used metric for the assessment of hydrological models, and is calculated as:

$$\text{NSE} = 1 - \frac{\sum_{t=1}^{T}\left(Q_t^\theta - \widetilde{Q_t}\right)^2}{\left(\widetilde{Q_t} - \overline{\widetilde{Q_t}}\right)^2} \tag{12}$$

NSE can range from $-\infty$ to 1, with NSE = 1 corresponding to perfect predictions of the observed data, and NSE < 0 indicates
the observed mean is a better predictor than the model. While limitations in the NSE have been identified (see, for example,





Gupta et al., 2009), the metric has been included as it provides a measure of performance for the hydrological model, and has interpretation of given the broad application of the metric in the literature.

### 3.7 Model Configurations under Consideration

Based on the methodology outlined, a number of model configurations have been compared: CRR models with and without state updating, and each calibrated to a rolling 10 year or 20 year window. Each configuration has been applied to the C1 and C2 catchments, and driven by both observed and forecast rainfall. Hence, there were 64 different cases considered in this work.

Two different influences of the calibration period on the models were analysed. The first was changes in the predictive distribution with changes in the length of data record used to calibrate the model parameters. The second was the rolling approach used to calibrate the model parameters, where the model was recalibrated at the start of the water year to the most recent data, which allowed model parameters to change over time. Each influence is considered further in the following section.

### 4 Results

The results for each of the cases considered for each of the metrics outlined above are summarised in Figure 4. The original values for each metric that correspond to the minimum and maximum bounds in Figure 4 can be seen in Table 2. From Figure 4, in general, the model that included state updating and was calibrated to the shorter period produced the best results across the metrics considered. The changes due adopting the state updating approach, for the different calibration periods, are considered in more detail below.

### 4.1 Impact of State Updating

The two options for initialising the model's routing store, a continuous 'warm up' period, or updating the routing store level based on the observed flow at the start of the forecast month, are shown as the red and green bars in Figure 4 (darker colours for the 10 year calibration period, and lighter colours for the 20 year calibration period).

The models with state updating improved the precision and bias in all cases considered (Figure 4). This can be seen in Figure 5, where the 90[th] percentile predictive limits for the models with and without state updating, and a 10 year calibration period and observed rainfall data, are presented for a subset of the record considered. From Figure 5 it can be seen that the predictive limits were generally more precise when state updating was adopted compared to without state updating, particularly during the periods when there was some flow occurring in the winter-spring months. However, in dry months, state updating can be seen to slightly widen the predictive limits (for example at the start of 2001 for C2), potentially capturing the low flows that occurred at this time more accurately.



State updating also improved the reliability metric in 7 of the 8 comparative cases, with the only exception being for C2 and the 20 year calibration period with forecast rainfall. As an example of the improved reliability from state updating, PQQ plots with and without state updating, for the 10 year calibration period observed rainfall cases, can be seen in Figure 6. A line for a given predictive distribution closer to the 1:1 line represents a more reliable distribution, and it can be seen this is the case

for the models that include state updating for both catchments. Any detrimental impacts on the predictive distribution from state updating tended to occur when the model storage was updated to represent a zero flow, and then a low flow that did occur in the following month was underestimated, e.g. in January 2002 for C2 in Figure 5.

## 4.2    Impact of Calibration Period

### 4.2.1    Differences in predictive distribution

The changes due to the different calibration period lengths can be seen by comparing the darker to lighter shades of each colour (darker colour 10 year period, lighter colour 20 year period) in Figure 4.

Most metrics improved for the shorter calibration period where state updating was not used. The reduction in performance for the longer calibration period could be due to the model being calibrated to data that represent higher yielding conditions from the past, where the catchment displayed a higher runoff coefficient (i.e. the start of the 1990s in Figure 3) than in the forecast

period.

However, where state updating was used, it was able to compensate for some of the changes, with smaller differences between the dark and light green bars compared to the dark and light red bars in Figure 4. This improvement is potentially due to the observed flow data being able to correct the model for any systematic overestimation in the simulated streamflow. The differences were more pronounced for the more practically relevant case with forecast rainfall, which introduced further errors

to be compensated for by the state updating approach, and the postprocessor error model.

One aspect of the impact of different calibration periods that was not obvious from the changes in values for the metrics was trends in the model outputs when compared to the observed streamflow over time. These trends are illustrated by the time series at the end of the period considered, presented in Figure 7.  As noted in Section 2.2, catchment C1 had been identified to have a substantial reduction in the rainfall-runoff relationship over time (Figure 3). From Figure 7, it can be seen that the model

calibrated to the longer period overestimated the observed flow in 2009 and 2010 in catchment C1, whereas the model calibrated to the shorter period provided a better reflection of the catchment response. In contrast, in catchment C2, which had a much more consistent relationship between rainfall and runoff over time in Figure 3, the differences in the 90th percentile predictive limits due to the calibration period in Figure 7 were minimal.



### 4.2.2 Differences in trends in parameter values

The approach of recalibrating the CRR model parameters each year to the preceding data relevant for the calibration period considered (i.e. the previous 10 or 20 year period) allows for any temporal trends in the parameter distributions to be investigated (see Section 3.2). Figure 8 presents the median, and 90th percentile prediction limits of these distributions for each
parameter for each catchment, for the 10 year and 20 year calibration periods, as different colours.

In catchment C1, for the first 10-12 years of the record, the median value for each parameter was similar for both calibration periods, with slightly wider bounds for the shorter calibration period, likely due to the reduced data available to infer representative parameter values. After this period, but particularly in the last 5 years, the shorter calibration period can be seen to identify values that result in reduced runoff, in particular more negative values for X2, the groundwater exchange coefficient.
Reducing values for X4, the time base of unit hydrograph, may appear a strange result because it corresponds to a modification of the travel time within the catchment. However, the reduced value of X4 may represent a reduced contributing catchment area through interception in the upper reaches of the catchment. Cross-correlation analysis of the rainfall-runoff data supports this reduction in travel time based on the lag between rainfall and runoff producing the maximum correlation value, when comparing between the start and end of the dataset (Gibbs et al., 2017).

In catchment C2, the parameter values resulting from the different calibration periods were similar to each other. The distinct change in parameter values for the 10 year calibration period in 1999 appears to be a model fitting anomaly resulting from a shorter calibration period. This finding highlights that longer calibration periods would generally help identify more robust parameter values. There were no substantial differences in parameter values for the calibration periods considered, which may be expected given the more consistent rainfall-runoff relationship over time (Figure 3). However, there were trends in the
parameter values over time for both calibration periods. Considering the results from the 20 year calibration period, the statistically significant trends ($p<0.05$) in the median values of the model parameters were $\Delta X1 = 3.96$ mm/year and $\Delta X3 = -5.17$ mm/year. Both trends are in the direction of parameters values that reduce runoff for a given amount of rainfall, through increased evaporation (increase in X1, maximum capacity of the production store) and increased loss to inter-basin transfers (reduction in X3, maximum capacity of the routing store).

## 5 Discussion

The state updating models were found to improve the predictive distributions produced, across the performance metrics for the catchments considered. State updating would be expected to reduce the simulated bias, as errors in the simulated streamflow during the warm up period are corrected at the start of the forecast period. State updating would be also expected to increase the precision of the predictive distribution, as the range in model predictions should be reduced by forcing the model to simulate
the observed streamflow at the start of the forecast period. However, the trade-off for an increase in precision would typically





be a reduction in the reliability of the predictive distribution (e.g. McInerney et al., 2017). This was not the case in this work, where in all but one of the cases considered, reliability also improved.

The state updating approach adopted could be modified to potentially further improve the reliability of the forecasts produced. For example, the approach could be extended to also update the GR4J production store along with the routing store. This could be expected to improve the results in the periods where detrimental changes were found, in months with very low flow and limited storage in the routing store to update. Given the main detrimental impacts of state updating occurred in the no/low flow period, the approach could also be adopted only in periods where there was some recorded flow. However, forcing the model to correctly represent an observed no flow may also be beneficial (for example, in late 2002 and into 2003 in Figure 5), as an approach such as this requires further investigation.

Out of context, the calibration period results could be seen to suggest that the shorter calibration period provided better (or at least not detrimental) parameter estimates. However, this interpretation would not be expected to be a general result, nor would it indicate that even shorter calibration periods would further improve predictions. Ultimately, our empirical findings highlight the benefits of identifying the longest period of data that is representative of conditions of interest for a given model application, which is a task often overlooked in practical application. The expected result that longer calibration periods improve parameter identifiability can be seen in Figure 8, however for some catchments, there may be a trade-off with calibrating a model to data that are no longer representative of the current or future rainfall-runoff regime. As such, data suitable for model calibration should be identified through trend analysis and other knowledge of changes in a catchment (e.g. land use data, abstraction volumes). The results presented indicate that if this step is ignored, and the modeller naively assumes that longer calibration periods are inherently better for model development, detrimental impacts to all aspects of the predictive distribution can occur.

## 6    Conclusions

This work investigated the ability to produce streamflow forecasts for an application in southern Australia, where streamflow can be diverted to support a range of environmental and social outcomes, and where improved information on future water availability can assist in decision support and management. As part of meeting this objective, several methods related to improving the predictive ability of streamflow forecasts were investigated.

The first main contribution of this study was the implementation of the state updating approach to assimilate recently observed streamflow into the conceptual rainfall-runoff model GR4J, and to assess the implications of this approach on the forecast predictions. The results indicate that this simple state updating approach could improve all aspects of the predictive distribution, including the precision and reliability of the forecasts.

The second main major contribution of this study was to establish the impact of the length of the calibration period on the forecast accuracy. The results indicated that the shorter calibration period improved model performance in most cases, contrary



to standard approach of using as much data as possible for model calibration. The reduction in performance for the longer calibration period is likely due to the model being calibrated to data that represent higher yielding conditions from the past, where the catchment displayed a higher runoff coefficient than in the forecast period. This result highlights that identifying a data set representative of the forecast period, through trend analysis and other knowledge of a catchment, is an important step

in model development. The results presented indicate that if this step is ignored, and it is assumed that the more data the better for model development, detrimental impacts to all aspects of the predictive distribution can result.

The conclusions drawn in this empirical study are limited by the one case study region considered. While the results are intuitive, further work should consider a larger sample of catchments, encompassing a range of flow regimes. Similarly, only one CRR model structure has been considered, and the state updating approach, and benefits thereof, may change for different

model structures. Further work should also consider the merits of updating other aspects of the models considered, for example similar or improved performance may be achieved through updating the error estimated by the postprocessor error model instead of, or as well as, the CRR model storage level. Nonetheless, this study has demonstrated a number of relatively simple techniques for pre-processing, post-processing and calibration approaches to enhance monthly streamflow forecasting skill.

**Author Contribution**

M. Gibbs performed the analysis and produced the manuscript, with contributions from all co-authors. H. Maier and G. Dandy assisted with the design of the project. D. McInerney undertook the postprocessor error modelling and analysis, with help from M. Thyer and D Kavetski. G. Humphrey implemented the climate model forecast downscaling to generate the inputs for the hydrological models.

**Acknowledgements**

The flow data used in this paper is available from the South Australian Department for Environment, Water and Natural Resources Surface Water Archive (https://www.waterconnect.sa.gov.au/Systems/swd). The climate data used in this paper is available from the Queensland Department of Science, Information Technology, Innovation and the Arts SILO climate data archive (https://www.longpaddock.qld.gov.au/silo/). Access to forecast climate data from the POAMA2 model was gratefully provided by the Bureau of Meteorology (http://poama.bom.gov.au/). M. Gibbs and G. Humphrey were supported by the

Goyder Institute for Water Research, Project E.2.4. D. McInerney was supported by Australian Research Council grant LP140100978 with the Australian Bureau of Meteorology and South East Queensland Water. Input from South East Water Conservation and Drainage Board staff, in particular Senior Environmental Officer, Mark DeJong, is gratefully acknowledged.





**Table 1 Bounds adopted for the uniform prior distribution**

| Parameter | Lower Bound | Upper Bound |
|-----------|-------------|-------------|
| X1 | 100 | 600 |
| X2 | -15 | 5 |
| X3 | 1 | 300 |
| X4 | 0.5 | 6 |
| split | 0.6 | 0.99 |

**Table 2 Minimum and maximum values for each metric across all models and catchments considered.  Higher values are better for CRPS$_{SS}$ and NSE, lower values are better for the other metrics. The values in this table should be interpreted alongside Figure 4, where the values here corresponding to the 0.05 and 0.95 relative values in Figure 4.**

| | Reliability | Precision | Bias | CRPS$_{SS}$ | NSE |
|---------|-------------|-----------|------|-------------|------|
| Minimum | 0.08 | 0.45 | 0.04 | -0.64 | -1.01 |
| Maximum | 0.39 | 2.18 | 1.48 | 0.65 | 0.91 |





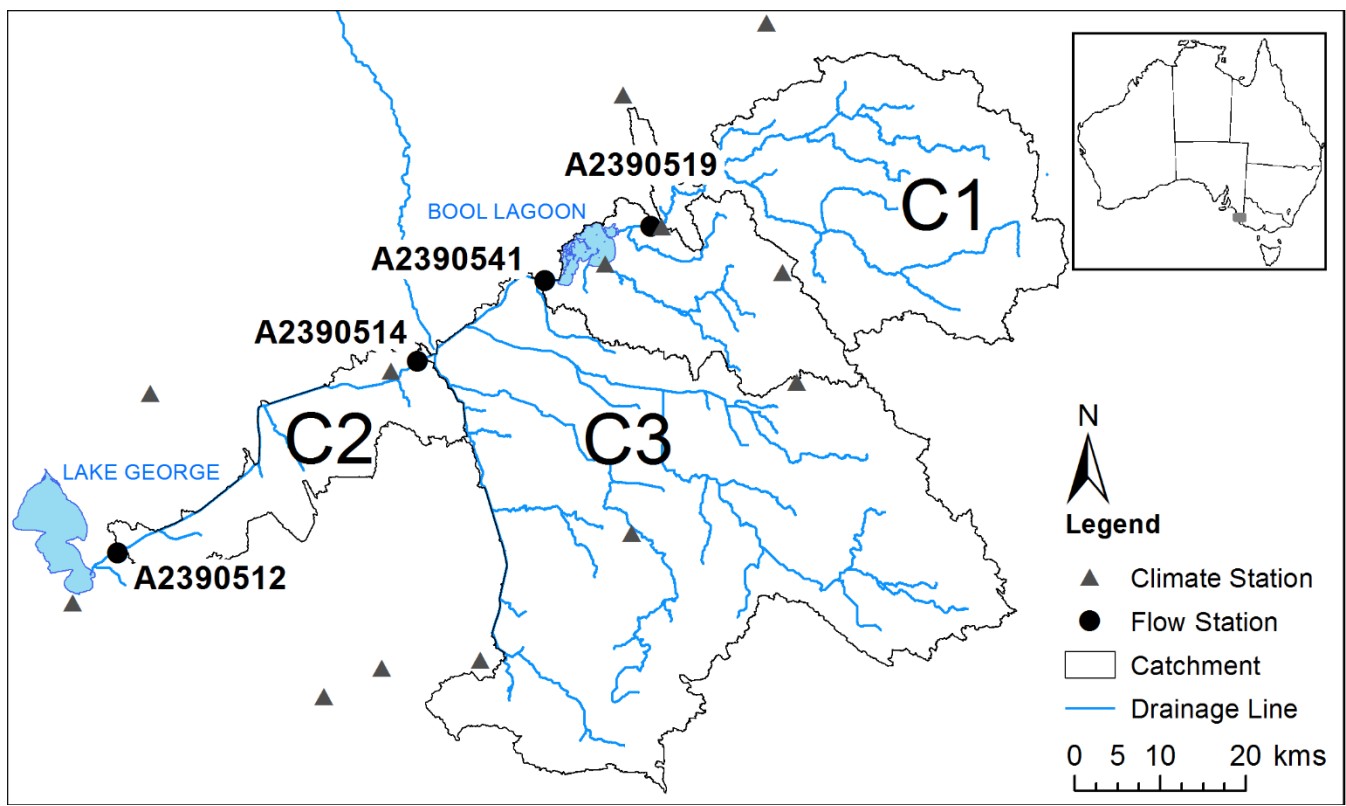

**Figure 1 Map of the forecasting application region, in the south east of South Australia**

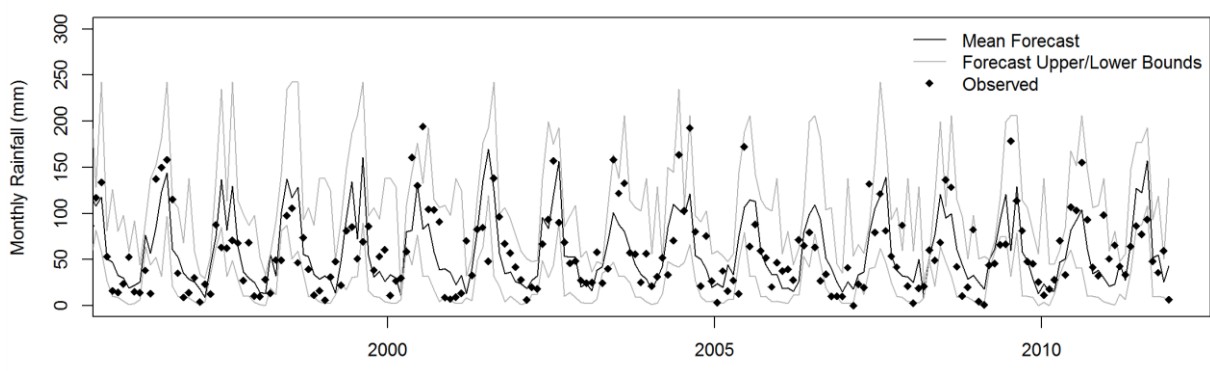

**Figure 2 Example of downscaled POAMA rainfall forecasts at station 26000 in comparison to observed rainfalls**





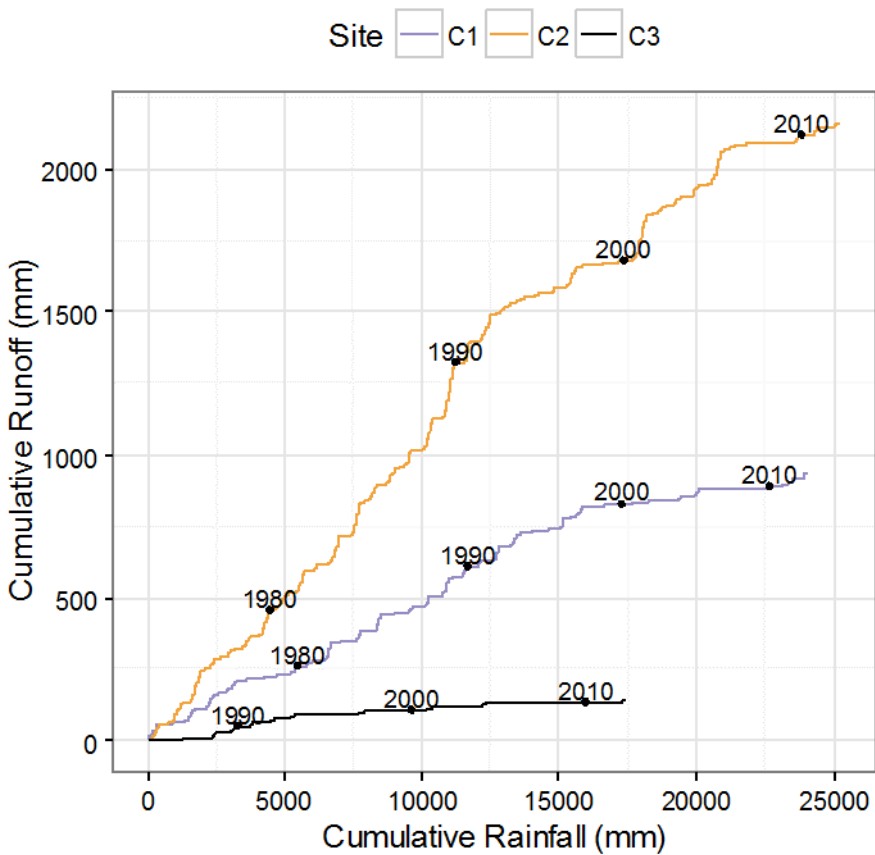

**Figure 3 Double mass plot of the three main catchments contributing to Drain M. It can be seen that 1) the volume of runoff for the same volume of rainfall has reduced in the latter decade, and 2) very little runoff is generated from the C3 catchment.**





Figure 4 Comparison of metrics across catchments (C1 and C2) and source of rainfall data. The performance of models with and without state updating can be seen between the red and green bars (comparing the darker shades and lighter shades). The change in performance due to different calibration periods (CP) can be compared between the darker and lighter shading (darker colour representing the 10 year period, and lighter colour 20 year period).



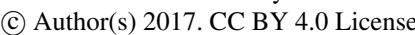

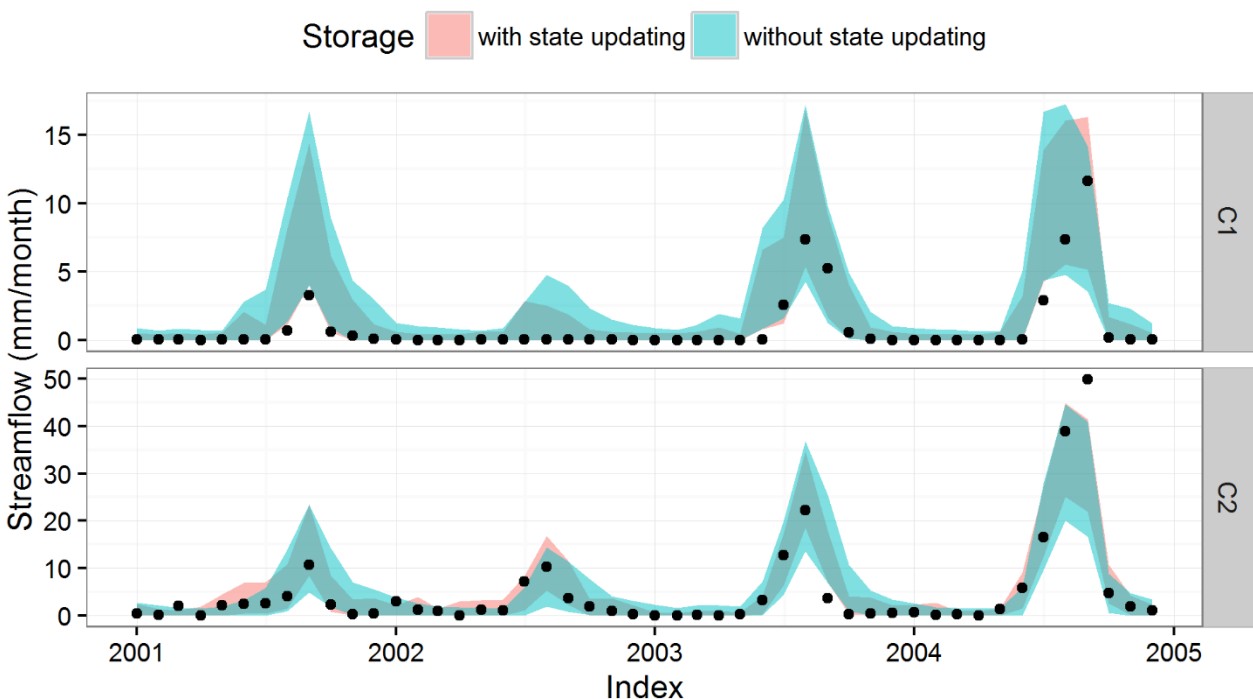

**Figure 5 Representative time series of the 90th percentile prediction limits, with observed values as black dots, for the 10 year calibration period observed rainfall models. State updating can be seen to increase precision compared to without state updating.**





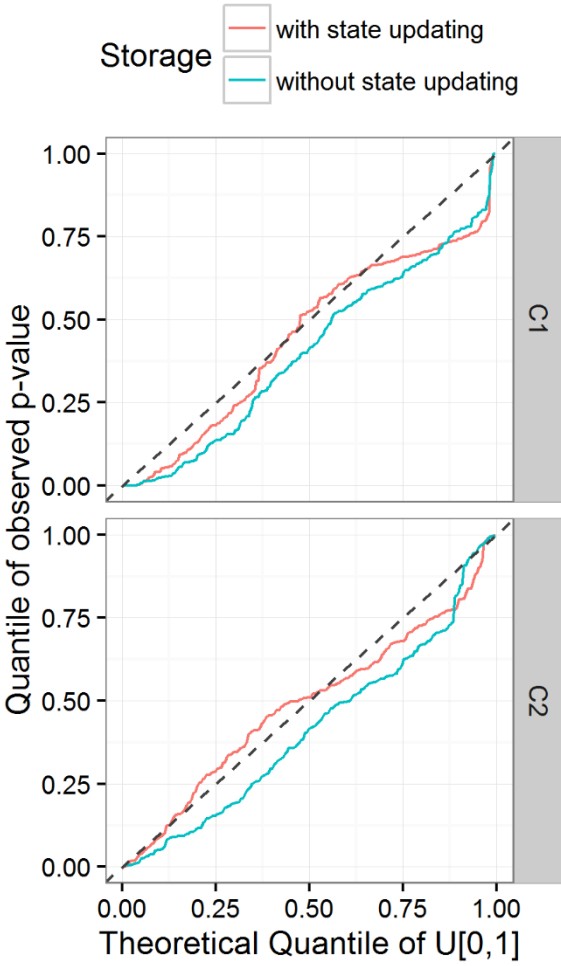

**Figure 6 Predictive Quantile – Quantile plots for models with the 10 year calibration period and forced by observed rainfall, comparing reliability with and without state updating. State updating can be seen to increase the reliability of the predictions.**





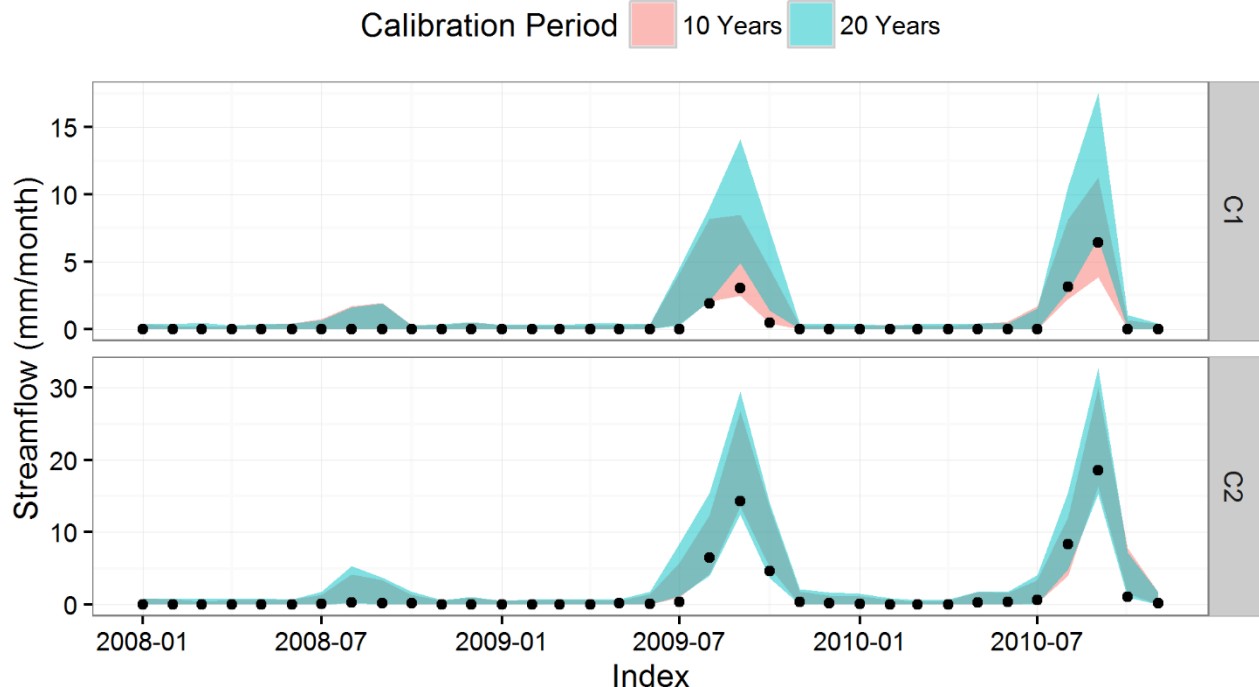

**Figure 7 Representative time series of the 90th percentile prediction limits, with observed values as black dots, for the models with state updating and observed rainfall. For the catchment with non-stationary changes (C1), the use of the longer calibration period results in overestimated flow at the end of the period, due to the influence of data earlier in the calibration period.**





**Figure 8 Temporal trends in posterior parameter distributions, for catchments C1 (top) and C2 (bottom). The median values are shown as the solid lines and the shaded area represent the 90[th] percentile prediction limits.**





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
