# Peer review of "State Updating and Calibration Period Selection to Improve Dynamic Monthly Streamflow Forecasts for an Environmental Flow Management Application"

_Hydrology and Earth System Sciences, 2017_

## Short Comment (SC1)

General:

The manuscript of Gibbs et al. evaluates the effect of calibration setup on the GR4J model performance within 2 Australian basins on 1-month lead-time hydrologic forecast. Authors draw mainly following conclusions based on 2-basin analysis using 5 indicators: (I) the length of calibration period does not necessarily should be as long as possible, in particular when changes in flow regimes are observed. Additionally, (II) the authors state that a simple model state updating improves hydrological forecast at 1-month lead time.

Based on my review, I consider the overall topic to be relevant for HESS, however, some parts of the manuscript need to improved and clarified, as further suggested below.

Thank you for the constructive comments that will improve the clarity and contribution of the manuscript.

Major comments:

1. In general, it is not surprising that updating model initial conditions has benefits on hydrologic forecast. Additionally, it is not surprising that changes of physiographic conditions may change the catchment's response, indicating different information content/validity of observed discharge data on model parameters. This only confirms observations of previous studies, which some of them are cited. Would be nice to more clearly demonstrate benefits over existing/operational approaches (in terms of costs etc). In particular, when the title includes words like "Management Application".

Commentary on the benefits over the existing approach will be added to the Discussion section. The approach currently used by the management authority is very conservative. Essentially forecasts are not considered and changes in water management are made only once downstream requirements have been met. With the forecasting models and methods developed, it is now possible to provide a probabilistic estimate of how likely it is that the downstream requirements will be met in the next month, to allow the risk in changing management actions earlier to be assessed. This is expected to improve the outcomes achieved from the volume of fresh water available, by enabling decisions on management changes to be made earlier in the season.

The Discussion section will be expanded to contrast the findings in this work against the relevant references cited (Brigode et al., 2013; Luo et al., 2012), to highlight where there are similar or different findings.

2. Unfortunately, the analysis is limited to two basins, which really can't be used to draw any conclusions (as authors also recognise in the end). I would strongly encourage authors to enlarge the number of basins and events. Another two basins may yield completely different results; therefore, generality should be avoided.

It is agreed that general results cannot be drawn from the results based on two basins. The drainage system considered was based on a user need for seasonal forecasts, which resulted in the limited application to two basins. Although the procedure has not been tested on a wide range of basins, the catchments selected are ephemeral and therefore are known to be more challenging to provide good quality forecasts (Ye et al., 1997). So even though only two basins have been considered, they are expected to be 'challenging' basins.

Future work will consider if, and under what conditions, the results for state updating at the monthly scale and effect of the length of the calibration period apply, beyond the case study considered. This point, on the limited generality of the results and need for further work on more basins, will be included in the Discussion section.

3. Would be nice to relate your results with another study, which details CRR state updating for 1-month forecast. However, I wonder, whether it is really the effect of state updating here. It is well recognized that the effect of initial conditions (based on discharge observations), in such small basins, diminishes after a couple of days. Please, comment on this.

The result that model updating still improved forecast performance at the lead-time of one month is considered one of the key results of this work. The controlled study for testing the model with and without state updating for all the cases considered (observed and forecast rainfall, two lengths for the calibration period and two basins), demonstrates that for the scenarios considered the effect can be attributed to state updating. The point at Comment 2, on the potentially limited generality of this result, is relevant and will be included.

It is agreed that the majority of the literature focused on short term flood forecasting has found limited effect of initial conditions after a few days (e.g. Berthet et al., 2009). However, forecasting of flood peak and timing is a different application to that considered here, forecasting water volume available for environmental use. A number of data driven modelling studies have demonstrated that monthly streamflow lagged by one (or more) months provided some useful information for forecasting a one month lead time (e.g. Bennett et

al., 2014; Humphrey et al., 2016; Yang et al., 2017). This difference in application, and relevant literature, will be included in the introduction of the revised manuscript.

The description of basin and data is way too long and detailed (3 pages), in particular when the discussion and conclusion do not come to those details at all.

Reviewer 1 raised a similar issue, and as such Section 2 will be shortened to be more targeted and remove surplus detail, such as that of the water balance model. One of the topics of interest for the sub-seasonal to seasonal hydrological forecasting special issue was user needs for seasonal forecasts. As such, more detail than typical on the case study application to wetland management was included in the manuscript. However, it is agreed that this is a distraction from the more generic scientific issues of interest to most readers. As outlined at Comment 1, commentary on the benefits over the existing approach will be added to the Discussion section.

4. Please, place error bars into figure 4, in the same way as the uncertainty is presented in following figures and provide discussion.

The uncertainty in the streamflow time series had been produced and assessed (e.g Figures 4 & 5) because it is a direct of the output of the modelling setup. However, all of the streamflow replicates are used in the calculation of the performance metrics (reliability, precision and CRPS in particular), and as such the uncertainty in the values of the metrics is not easily derived. To the authors' knowledge, it is no common to provide the uncertainty in performance metrics related to streamflow forecast uncertainty (e.g. Alfieri et al., 2014; McInerney et al., 2017; Pappenberger et al., 2015; Renard et al., 2010).

We do acknowledge that there is uncertainty in the values for the metrics, due to finite number of replicates and the finite length of the observed data (McInerney et al., 2017) and that in the presence of strong autocorrelation in the streamflow error time series this uncertainty may be significant. However, to develop an approach to quantify the metric uncertainties is beyond the scope of this study. Instead, we follow the guidance provided by McInerney et al. (2017) who suggested that these uncertainties are quite small and are unlikely to impact on the conclusions of the study. We do acknowledge that further work is needed in this area.

6. Discussion about alternative types of observations (besides Q) may be provided in the manuscript.

The literature review will be expanded to refer to previous studies that have used other types of observations that could be used for state updating (e.g. either remotely sensed or ground based soil moisture), and why streamflow was focused on in this study.

7. What is the main applicability of your findings? Are they going to be used operational, if yes, what are the benefits over existing forecast method? Please, clarify.

See response to Comment 1.

Minor:

- Third sentence from the Introduction regarding Drain M catchment should be moved somewhere towards the end of Introduction.

This change will be made.

- P 6, L.5: evapoconcentration => evapotranspiration?

This typo will be corrected

- Section name 2.2 "Streamflow and streamflow data": sounds a bit repetitive

The section will be changed to "Streamflow data"

- P 9, L.21 "burn-in" into quotes

This change will be made.

- Eq. 12 is wrong, sum in the denominator is missing

Correct, the equation will be updated.

- Caption of figure 2: "POAMA" => "POAMA-2"

This change will be made.

**References**

Alfieri, L., Pappenberger, F., Wetterhall, F., Haiden, T., Richardson, D., and Salamon, P.: Evaluation of ensemble streamflow predictions in Europe, Journal of Hydrology, 517, 913-922, 2014.

Bennett, J. C., Wang, Q. J., Pokhrel, P., and Robertson, D. E.: The challenge of forecasting high streamflows 1 & 3 months in advance with lagged climate indices in southeast Australia, Nat. Hazards Earth Syst. Sci., 14, 219-233, 2014.

Berthet, L., Andreassian, V., Perrin, C., and Javelle, P.: How crucial is it to account for the antecedent moisture conditions in flood forecasting? Comparison of event-based and continuous approaches on 178 catchments, Hydrology and Earth System Sciences, 13, 819-831, 2009.

Brigode, P., Oudin, L., and Perrin, C.: Hydrological model parameter instability: A source of additional uncertainty in estimating the hydrological impacts of climate change?, Journal of Hydrology, 476, 410-425, 2013.

Humphrey, G. B., Gibbs, M. S., Dandy, G. C., and Maier, H. R.: A hybrid approach to monthly streamflow forecasting: Integrating hydrological model outputs into a Bayesian artificial neural network, Journal of Hydrology, 540, 623-640, 2016.

Luo, J., Wang, E., Shen, S., Zheng, H., and Zhang, Y.: Effects of conditional parameterization on performance of rainfall-runoff model regarding hydrologic non-stationarity, Hydrological Processes, 26, 3953-3961, 2012.

McInerney, D., Thyer, M., Kavetski, D., Lerat, J., and Kuczera, G.: Improving probabilistic prediction of daily streamflow by identifying Pareto optimal approaches for modeling heteroscedastic residual errors, Water Resources Research, 53, 2199-2239, 2017.

Pappenberger, F., Ramos, M. H., Cloke, H. L., Wetterhall, F., Alfieri, L., Bogner, K., Mueller, A., and Salamon, P.: How do I know if my forecasts are better? Using benchmarks in hydrological ensemble prediction, Journal of Hydrology, 522, 697-713, 2015.

Renard, B., Kavetski, D., Kuczera, G., Thyer, M., and Franks, S. W.: Understanding predictive uncertainty in hydrologic modeling: The challenge of identifying input and structural errors, Water Resources Research, 46, 2010.

Yang, T., Asanjan, A. A., Welles, E., Gao, X., Sorooshian, S., and Liu, X.: Developing reservoir monthly inflow forecasts using artificial intelligence and climate phenomenon information, Water Resources Research, 53, 2786-2812, 2017.

Ye, W., Bates, B. C., Viney, N. R., Sivapalan, M., and Jakeman, A. J.: Performance of conceptual rainfall-runoff models in low-yielding ephemeral catchments, Water Resources Research, 33, 153-166, 1997.

---

## Referee Comment (RC1) · Anonymous Referee #1 · 10 Aug 2017

August 2017, the 9th

A review of the article entitled "State Updating and Calibration Period Selection to Improve Dynamic Monthly Streamflow Forecasts for a Wetland Management Application", submitted to HESS by Gibbs et al.

This article investigates 2 scientific issues in the context of rainfall-runoff seasonal forecasting: (a) the advantages of state(s) updating and (b) the sensitivity of the choice of the data used for calibration (calibration period length). These 2 scientific issues

have been / are widely discussed in the hydrological community. Authors choose to put them in the context of complex wetland management application. This intention is relevant, since these issues are of crucial importance for operational matters.

First, it is worth noting that the manuscript is most often very clear (in particular, the introduction is efficient). Some suggestions are made below (detailed comments) to make the manuscript clearer (some parts are easier to understand when checked again after a further reading). The methodology is quite well detailed (I reckon that it is sufficient for anyone who wishes replicating the study) and the results are well presented. However, many (too many ?) details concerning the application context are provided (section 2). I am afraid that I missed understanding how they infer with the scientific issues: results and discussion section do not make clear to me whether and how this particular context has implication on the way these issues are dealt with and on the results of the study. In a similar way, some details are given about the data used in an operational context, but it is not clear how they impact the results of this study. For example, this is the case of the precipitation forecasts (see detailed comments). Indeed all the results are not discussed in depth, with respect to these options (e.g., results obtained with observed rainfall versus results obtained with forecasted rainfall) and with respect to the context and practical purposes of the wetland management (whereas this appears in the submitted title): results are presented in only 2 pages and a half and the discussion is shorter (1 page). Even if all the tested cases are very useful for the specific case study and application, they are then not fundamental for the reader who focuses more on the 'generic' scientific issues than on the specific context of wetland management. The authors should consider removing them in order to make the reading and the analysis easier, rather than providing everything they learnt from their case study (again: even if it is quite interesting per se). They may prefer explaining how their findings are related to their specific case study and practical application.

As mentioned previously, the 2 scientific issues have been explored by many previous studies. That is why this article has to do thorough review of literature in order to

emphasize on the novelty of their study or to compare their results to those of other studies:

- The way how non-stationarity is treated is very satisfying. The explicit distinction between physical catchment non stationarity and other model non-stationarity is necessary (while not always made); it is introduced in a very clear manner. I only suggest the authors to give a more explicit definition of the model parameters (the discussion is indeed implicitly present behind), since some previous studies proposed parameter variations to compensate many different non-stationarities (up to model structural deficiencies), as nicely pointed out in the introduction. This issue is particularly relevant in this study because the authors chose to update their model but only selected state updating, while many other approaches exist, one of them being parameter updating: this choice, which is very consistent, may be better explained. While not being a specialist of the choice of data calibration, I found the quoted references relevant. I only wish that these articles (e.g., Luo et al., 2011, which clearly inspired the methodology adopted by Gibbs et al.) would have been quoted not only in a generic way but also in sections 4 (Results) and 5 (Discussion) as benchmarks for the results: the results confirm previous studies in a large part; is there any interesting difference?

- Concerning the data assimilation and model updating issue, the bibliography is poorer (see detailed comment for page 3). Many references could be added. Since the authors chose to use the GR4J model and since the state updating they chose is the same one as the approach adopted for the GRP model ('adaptation' of the GR4J model for forecasting, used by the French flood forecasting centres), it is also worth mentioning this work (see detailed comments below). Beyond the references issue, it may (should ?) be noted that this study explores the benefits of model updating for seasonal forecasting, whereas many, if not most, studies consider shorter lead-times. This aspect has to be mentioned, since it is well known that the effects of model updating most often vanish whe the lead-time increases. In my opinion, keeping benefits at large lead-time is one of the (surprising) key result of this study and may be usefully emphasized.

[Figure]

A few methodological choices may deserve a little more discussion or explanation: - The calibration algorithm is a rather complex one, but used with assumptions which are known to be not met in most cases (page 9, line 7: independent, homoscedastic residuals). Moreover, these assumptions are not consistent with the choice of the model error post-processor (a Box-Cox transformation is used in order to take into account the heteroscedasticity of these same residuals). Why did the authors pick a complex approach with unverified and inconsistent assumptions rather than a simpler one? It let the reader think that the authors used "components" available on the shelf or a pre-existing tool, which is quite understandable. But then they have to justify these choices (and why a so complex calibration method when much simpler ones are easily available?). - Furthermore, one point is not discussed but may deserves some attention. Like the hydrological model, the model error post-processor is calibrated (not in a joint manner however). Why does the study on the impact of the calibration data period length on the calibration only focus on hydrological parameters and not on the post-processor parameters as well (mu, sigma)? This can indeed be treated independently (therefore not necessary in this article), but this research issue may be usefully mentioned. Are the post-processor parameters concerned by the rolling calibration?

One element may also be better detailed: the results are given at a monthly scale (time step), whereas the GR4J model is a daily one: the way the GR4J model is run has to be precised. This is important, since the model is updated and effects of model updating decrease when the lead-time increases. However, it often does not only depend on the lead-time 'absolute' value but also on the number of time steps to reach this lead-time.

In a nutshell, this article brings some interesting results, even in a field explored by many previous studies, and deserves publication. The suggestions made in order to improve the manuscript lead me to propose a moderate to major revision (however another round of submission afterwards does not seem necessary).

DETAILED COMMENTS

- Page 2

Line 22 ("As these models are conceptual, they require calibration [...]"): they are not the only models that do so. Even the (so-called) physically-based models which could theoretically not need calibration, are most often calibrated, for various practical reasons.

Lines 22 - 23: Brigode et al. (2013) show that this general a priori (longer calibration periods produce more robust parameters estimates) is not always verified. Therefore if this article is quoted (and it should be, in my opinion), it would be fair to indicate their results.

Lines 30 - 31: the definition of catchment non-stationarity which is proposed, is very interesting since it 'focuses' (restrains to) the physical object non-stationarity. However, it seems to be a binary state: the catchment is or is not stationary. Have the authors considered the notion of a "degree" of non-stationarity? Indeed, all the listed factors of non-stationarity are not expected to have the same consequences over the catchment behaviour. Might the rolling calibration approach be a tool to assess the relationship between "degrees" of non-stationarity and parameters evolution? (see also detailed comment on Fig. 3)

- Page 3

Line 1: is groundwater depletion a physical change (of the catchment) or the consequences of some of the listed catchment changes?

Lines 15 - 20: the bibliography review is rather poor: it gives some "extreme approaches" between the (too) simple GLUE and the very detailed BATEA (or similar approaches). Furthermore, GLUE is a quite old approach, giving a reference of 2008 is a bit strange (unfair?), as it appears more recent than much more advanced and sophisticated approaches, as those developed by Kavetski, Vrugt and others. Since the chosen approach is a model error post-processor, I suggest Krzysztofowicz and

Maranzano (2004).

Line 17: "using a model error post-processor" rather than "using a post-processor error model"?

- Page 4

Line 3 ("up to one month"): I understood this paragraph as a bibliography review giving general results (not specific to some catchments). However, it gives some values of the "influence duration" of the initial state, which strongly depends on the catchment characteristics. I am pretty confident in the fact that it easy to find catchments where the impact of the initial conditions is important during several months (even years).

Line 5: "warm-up" rather than "warmup"?

Lines 14-16: it may be specified that this impact has been deeply evaluated for shorter lead-times (this emphasizes the character of novelty of the study). Furthermore, I disagree with the second sentence as it has been shown that the impact decreases quite fast (for many not too slow catchments) and is almost negligible at a seasonal scale (see e.g. Berthet et al. 2009 that the authors quote elsewhere). That is one very interesting aspect of the results of the submitted study.

Line 18: I suggest to precise "calibration periods choice" or "calibration periods length" rather than only "calibration periods".

Line 21: to enhance "seasonal" forecasting skill?

Line 22: Does the article "demonstrate" that calibration period choice can affect forecast skill (that is quite known) or does it assess how much it does so?

- Page 5

Line 11: is it the gauge "A2390514" rather than "A21390514"?

Lines 26 - 27: an hydrograph may be useful to support this information.

[Figure]

- Page 6

Lines 3-13: is the description of the model developed by eWater Source useful for the reader. If I understood correctly, it is not directly related to the model used in this study. If so, this might confuse a bit the reader. E.g., I am not sure that the assumption of a constant inflow of salinity (which is not discussed) is needed by the reader to understand how the authors worked to answer to the scientific questions (which are the core of the article). The multi-objective nature of the calibration is also of no use for the rest of the study. If the fact that this model is used in practice had consequences on the methodological choices for this study, then the authors may consider explaining it (and discuss results with respect to it and to the specific context of wetland management).

Line 14 ("To use this model for to inform operations"): it is always tricky for a non native English speaker to ask so to native ones, but may the authors check English here?

Line 15: "lead-time" rather than "leadtime"?

Line 15: to fully understand the implication of the choice of the 1-month lead-time, it is necessary to know that the CRR model is a daily one (not only because the model is updated). However this information is given at subsection 3.1 (and not very explicitly: the reader has to know that GR4J is a daily model)

Line 19 ("reasonable forecast skill is expected to be possible compared to longer forecast horizons"): may the authors provide some references? How much are the performances expected to decrease for longer lead-times? Furthermore, why did the authors choose to focus on a single lead-time? The evolution of the benefits of the model updating, with respect to the lead-time, in a context of seasonal forecast, would be a very interesting result.

Line 20 ("The mean annual rainfall for the region is in the range 600-675 mm"): page 5, lines 3 and 4 suggest some spatial variability. How strong is it? (600 to 675 mm is not very strong difference, compared to some other climates around the globe).

Line 20: is it useful to precise what "FAO56" stands for?

- Page 7

Lines 3 - 4 ("2 rainfall hindcasts [...] were downscaled to the single rainfall gauge scale"): just to be sure, does it mean to the pixel where the gauge is?

Lines 15 - 25: the authors may consider whether this paragraph would not be better written earlier (e.g. among the first paragraphs of section 2).

Line 20 ("It should also be noted that releases from Bool Lagoon [...]"): why is it important to understand this scientific study? (I worry about missing something useful for the interpretation of the results)

Lines 30 and following: are the details about the streamflow measurements devices useful?

- Page 8

Line 3: I agree with the fact that indicating the data are of good quality and too often not done, but if the authors want to demonstrate the quality of the rating curves, they may add some information about the number of years during which the 78 and 166 gaugings have been achieved and how much often the rating curves have been modified.

Lines 9 - 20: since catchment non-stationarity is an important issue for this study, I suggest to make this paragraph a subsection dedicated to this topic (here).

Lines 23 - 24 ("GR4J [...] explicitly accounts for non-conservative (or 'leaky') catchments"): I agree. However, it should be kept in mind that GR4J has not been designed nor is known to achieve good performances for karstified catchments (mentioned page 7, line 13). Moreover, I am not convinced it is quite appropriate for ephemeral catchments (as suggested by line 21, page 11).

Lines 28 - 31: may the authors explain what motivates their choice of adding a 5th free parameter to calibration? Is it important for their particular catchments or for their

methodology in this study?

- Page 9

Lines 10 - 12 ("this function [RMSE] provides a focus on the highest flow in the time series, where the majority of the runoff occurs"): it is not necessary. It provides a focus on the largest absolute errors, which indeed most often occur for the largest flows. However, consider a hypothetical model whose errors would be only on low flows.

Line 26 ("External influences include model structural limitations [...]"): this confused me, after reading the catchment non-stationarity given on pages 2 & 3. Does it suggest that parameters variation due to structural deficiencies would be considered here?

- Page 10

Lines 3-5: Would not it be useful to emphasize the trade-off between a longer calibration period to reduce the parameter uncertainty and a shorter calibration period to mainly take into account the most recent dynamics in the introduction section?

Line 13-14: the literature review is also poor about data assimilation and model updating. Generic references may be Refsgaard (1997) and Liu and Gupta (2007). Since the chosen updating approach is the same as the one used for the GRP model (which is a mere adaptation of GR4J for forecasting purposes), I suggest to refer to the work of the team which developed these models. The authors may pick Tangara (2005) and Berthet (2010), both in French, which described the numerous tests of different updating approaches made by the GR4J research team (some of them discussed in section 5! See comment below) and detailed the resulting GRP model. They may prefer Berthet et al. (2010), which provides a much shorter description of the model and the updating techniques but also discusses the impact of the largest errors on the RMSE-based criteria values (see discussion page 9). For a detailed description of the GRP model, the authors may also consult: https://webgr.irstea.fr/en/modeles/modele-de-prevision-grp/fonctionnement-grp/. Moreover, since sequential approaches such as

ensemble Kalman filter and particle filters are mentioned, I suggest also to add references to Moradkhani et al. (2005, 5005b) and Weerts and El Serafy (2006).

Lines 17 - 27: a flowchart would greatly help the reader.

Line 27: "where X3 is the estimated runoff model parameter" rather than "where X3 is an estimated runoff model parameter"?

- Page 11

Lines 3 - 4 ("particularly when used to update both model state variables and model parameters"): I agree with the authors, but is it relevant here? (since parameters are not updated here).

Line 12 ("Depending on the case"): this is not clear, until the reader reaches section 3.7.

- Page 12

Line 5: why do the authors prefer to sample the (normalized) residuals rather than picking a number of calculated quantiles (from the Gaussian distribution)?

Line 12 ("Depending on the case"): this is not clear, until the reader reaches section 3.7.

Line 18: may the authors explain the choice of the 0.05 and 0.95 as normalized extrema values?

- Page 13

Lines 17 - 21: as pointed out by the authors, the reference distribution has an 'unfair' advantage. Then may the authors explain this choice? Why have they not chosen a simple naive forecast model?

Line 23: check the formula. There is missing sum for the denominator.

- Page 14

[Figure]

Line 6: is the rainfall forecast used here the ensemble forecasts described in subsection 2.1? I don't think it obvious. If not, why were the ensemble described?

Line 6: I found only 2^4 = 16 cases (model with or without updating; 2 calibration period lengths; 2 catchments and 2 rainfall forcings). What do I miss?

- Page 15

Line 6 ("Any detrimental impacts"): check English.

Line 8 (and followings): I suggest to precise "calibration period length" rather than only "calibration period"

Line 16: the differences are not much smaller for catchment C1. How much are they significant?

Line 19 ("the differences were more pronounced fo the most practically relevant cases with forecast rainfall [...]"): this is of particular interest for operational purposes (e.g., forecasts) and is worth being emphasized.

Line 20: I don't understand how the model error post-processor compensates the introduced errors. I thought that it only assesses them.

Lines 23-24 ("catchment C1 had been identified to have a substantial reduction in the rainfall-runoff relationship over time"). As discussed below (comments on Fig. 3), the catchment appears as rather stationary up to 1990 (approximately) and then also more or less stationary from 1990 to 2010. If it is so, how can the difference in calibrated parameters obtained with the 2 different calibration period lengths for years 2009 - 2010 be explained by this change around 1990?

- Page 16

Line 16 ("A model fitting anomaly resulting from a shorter calibration period"): did the author investigate this "anomaly"? How can it be explained?

[Figure]

Line 26: this is interesting at a seasonal scale, since it has been shown that hydrological models are "stable", i.e. the updating effect vanishes after a number of time steps (e.g. Berthet et al. 2009 at a hourly time step: then after a few days at most for a large majority of the tested watersheds).

Line 29 ("As the range in model predictions should be reduced by forcing the model to simulate the observed streamflow at the start of the forecast period"). Is there any confusion between precision and sharpness? Model updating increase sharpness and precision (at least for the shortest lead-times).

Line 30 ("the trade-off for an increase in precision would typically be a reduction in the reliability of the predictive uncertainty"): again I assumed that the authors meant "sharpness". I suggest to write that this trade-off for an increase in sharpness \*\* may \*\* result in the reduction of the reliability, if the authors do not provide a general (theoretical) explanation. As much as I know, this is a common feature, but exceptions should exist, and the last sentence of the paragraph (page 17, lines 1 - 2) says so.

- Page 17

Line 4 ("update the GR4J production store along with the routing store"): this has been tested by the GR4J team (Berthet, 2010), with no significant improvement in a forecasting context at a hourly time step.

Lines 4 - 5 ("This could be expected"): may the authors give some explanation to found this idea?

Lines 3 - 9: this paragraph is very interesting, but how is it related to the scientific issues developed in this article?

Lines 16 - 19: in my opinion, this is the (or one of the) key findings. How may the authors emphasize it, rather than putting it at the very end of the article?

Line 30 ("in most cases"): the study was driven only on 2 catchments... It should be pointed out that a work over a (much) larger number of catchments is needed to ensure

the generality of these interesting results.

- Page 19

Tab. 1: the authors may usefully add the parameters meanings and their units (and a GR4J flowchart aside).

- Page 20

Fig. 1: the drain M is not given in the legend and has the same color as catchment boundaries, which makes it difficult to identify.

Fig. 2: what are the upper and lower bounds? Minimum and maximum of the ensemble? Some predictive quantiles such as 0.05 and 0.95? If the latter, is there any information about the reliability?

- Page 21 (Fig. 3)

I wonder if there is not a "sudden" change around 1990: catchments C1 and C2 look quite stationary up to 1990, and also after 1990. If it is so, can it really be explained by plantation forestry expansion (which is more a "continuous" factor of non-stationarity). Furthermore, can this question the relevancy of the rolling calibration which might be better adapted for smooth non-stationarity?

- Page 22 (Fig. 4)

It is very important to insist on the fact that the plot gives relative values of the metrics (higher is better) which are then not consistent with the formulas and details given in subsection 3.6. I was first confused because I did not notice (at first) the mention in the y-label (even if I admit that it is written in (sufficiently) large characters). I suggest to change the criteria described in section 3.6 to give only their relative values.

- Page 23 (Fig. 5)

Why are the results plotted only up to 2005?

- Page 25 (Fig. 7)

Why are the results plotted only from 2008 to 2010?

REFERENCES

Berthet, L. (2010). Prévision des crues au pas de temps horaire : pour une meilleure assimilation de l'information de débit dans un modèle hydrologique. Ph. D. Thesis, IRSTEA (Antony), AgroParisTech (Paris), https://pastel.archives-ouvertes.fr/pastel-00529652v1

Berthet, L., Andréassian, V., Perrin, C. and Loumagne, C. (2010). How significant are quadratic criteria? Part 2. On the relative contribution of large flood events to the value of a quadratic criterion. Hydrological Sciences Journal, 55(6): 1063-1073. DOI: 10.1080/02626667.2010.505891

Krzysztofowicz, R. and Maranzano, C. (2004). Hydrologic uncertainty processor for probabilistic stage transition forecasting Journal of Hydrology, 293, 57-73

Liu, Y. and Gupta, H. (2007) Uncertainty in hydrologic modeling: Toward an integrated data assimilation framework. Water Resources Research, 43, W07401

Moradkhani, H., Sorooshian, S., Gupta, H. and Houser, P. (2005). Dual state-parameter estimation of hydrological models using ensemble Kalman filter. Advances in Water Resources, 28, 135-147

Moradkhani, H., Hsu, K.-L., Gupta, H. and Sorooshian, S. (2005b) Uncertainty assessment of hydrologic model states and parameters: Sequential data assimilation using the Particle Filter. Water Resources Research, 41, 1-17

Refsgaard, J. C. (1997). Validation and Intercomparison of Different Updating Procedures for Real-Time Forecasting Nordic Hydrology, 28, 65 - 84

Tangara, M. (2005). Nouvelle méthode de prévision de crue utilisant un modèle pluie-débit global. Ph. D. Thesis, IRSTEA (Antony) and École pratique des hautes études de

Paris, https://webgr.irstea.fr/wp-content/uploads/2012/07/2005-TANGARA-THESE.pdf

---

## Referee Comment (RC2) · Anonymous Referee #2 · 28 Aug 2017

General:

The manuscript of Gibbs et al. evaluates the effect of calibration setup on the GR4J model performance within 2 Australian basins on 1-month lead-time hydrologic forecast. Authors draw mainly following conclusions based on 2-basin analysis using 5 indicators: (I) the length of calibration period does not necessarily should be as long as possible, in particular when changes in flow regimes are observed. Additionally, (II) the authors state that a simple model state updating improves hydrological forecast at 1-month lead time.

[Figure]

Based on my review, I consider the overall topic to be relevant for HESS, however, some parts of the manuscript need to improved and clarified, as further suggested below.

Major comments:

1. In general, it is not surprising that updating model initial conditions has benefits on hydrologic forecast. Additionally, it is not surprising that changes of physiographic conditions may change the catchment's response, indicating different information content/validity of observed discharge data on model parameters. This only confirms observations of previous studies, which some of them are cited. Would be nice to more clearly demonstrate benefits over existing/operational approaches (in terms of costs etc). In particular, when the title includes words like "Management Application".

2. Unfortunately, the analysis is limited to two basins, which really can't be used to draw any conclusions (as authors also recognise in the end). I would strongly encourage authors to enlarge the number of basins and events. Another two basins may yield completely different results; therefore, generality should be avoided.

3. Would be nice to relate your results with another study, which details CRR state updating for 1-month forecast. However, I wonder, whether it is really the effect of state updating here. It is well recognized that the effect of initial conditions (based on discharge observations), in such small basins, diminishes after a couple of days. Please, comment on this.

4. The description of basin and data is way too long and detailed (3 pages), in particular when the discussion and conclusion do not come to those details at all.

5. Please, place error bars into figure 4, in the same way as the uncertainty is presented in following figures and provide discussion.

6. Discussion about alternative types of observations (besides Q) may be provided in the manuscript.

7. What is the main applicability of your findings? Are they going to be used operational, if yes, what are the benefits over existing forecast method? Please, clarify.

Minor:

• Third sentence from the Introduction regarding Drain M catchment should be moved somewhere towards the end of Introduction.

• P 6, L.5: evapoconcentration => evapotranspiration?

• Section name 2.2 "Streamflow and streamflow data": sounds a bit repetitive

• P 9, L.21 "burn-in" into quotes

• Eq. 12 is wrong, sum in the denominator is missing

• Caption of figure 2: "POAMA" => "POAMA-2"

---

## Author Comment (AC1) · 30 Aug 2017

This article investigates 2 scientific issues in the context of rainfall-runoff seasonal forecasting: (a) the advantages of state(s) updating and (b) the sensitivity of the choice of the data used for calibration (calibration period length). These 2 scientific issues have been / are widely discussed in the hydrological community. Authors choose to put them in the context of complex wetland management application. This intention is relevant, since these issues are of crucial importance for operational matters.

First, it is worth noting that the manuscript is most often very clear (in particular, the introduction is efficient). Some suggestions are made below (detailed comments) to make the manuscript clearer (some parts are easier to understand when checked again after a further reading). The methodology is quite well detailed (I reckon that it is sufficient for anyone who wishes replicating the study) and the results are well presented.

However, many (too many ?) details concerning the application context are provided (section 2). I am afraid that I missed understanding how they infer with the scientific issues: results and discussion section do not make clear to me whether and how this particular context has implication on the way these issues are dealt with and on the results of the study. In a similar way, some details are given about the data used in an operational context, but it is not clear how they impact the results of this study. For example, this is the case of the precipitation forecasts (see detailed comments). Indeed all the results are not discussed in depth, with respect to these options (e.g., results obtained with observed rainfall versus results obtained with forecasted rainfall) and with respect to the context and practical purposes of the wetland management (whereas this appears in the submitted title): results are presented in only 2 pages and a half and the discussion is shorter (1 page). Even if all the tested cases are very useful for the specific case study and application, they are then not fundamental for the reader who focuses more on the 'generic' scientific issues than on the specific context of wetland management. The authors should consider removing them in order to make the reading and the analysis easier, rather than providing everything they learnt from their case study (again: even if it is quite interesting per se). They may prefer explaining how their findings are related to their specific case study and practical application.

One of the topics of interest for the sub-seasonal to seasonal hydrological forecasting special issue was user needs for seasonal forecasts. As such, more detail than typical on the case study application to wetland management was included in the manuscript. However, it is agreed that this is a distraction from the more generic scientific issues of interest to most readers. As such, section 2 will be shortened to be more targeted and remove surplus detail, such as that of the water balance model. See responses to detailed comments below for further changes in this regard.

It is believed that all the test cases considered remain relevant to the contribution of the manuscript and general scientific issues, i.e. with and without state updating, two different calibration period lengths, and observed and forecast rainfall. The least contribution may come from the two rainfall sources, but these cases represent two different tests. The observed rainfall allow the effect of the changes on model performance to be isolated, without the effects and errors associated with the forecast rainfall propagating through. However, the forecast rainfall case is necessary for the practical application, and without this case the results cannot be considered realistic.

As mentioned previously, the 2 scientific issues have been explored by many previous studies. That is why this article has to do thorough review of literature in order to emphasize on the novelty of their study or to compare their results to those of other studies:

-        The way how non-stationarity is treated is very satisfying. The explicit distinction between physical catchment non stationarity and other model non-stationarity is necessary (while not always made); it is introduced in a very clear manner. I only suggest the authors to give a more explicit definition of the model parameters (the discussion is indeed implicitly present behind), since some previous studies proposed parameter variations to compensate many different non-stationarities (up to model structural deficiencies), as nicely pointed out in the introduction. This issue is particularly relevant in this study because the authors chose to update their model but only selected state updating, while many other approaches exist, one of them being parameter updating: this choice, which is very consistent, may be better explained. While not being a specialist of the choice of data calibration, I found the quoted references relevant. I only wish that these articles (e.g., Luo et al., 2011, which clearly inspired the methodology adopted by Gibbs et al.) would have been quoted not only in a generic way but also in sections 4 (Results) and 5 (Discussion) as benchmarks for the results: the results confirm previous studies in a large part; is there any interesting difference?

A description of the model parameters and structure will be included in the revised manuscript, see response to detailed comments below. The rolling approach used to calibrate the model parameters (both the hydrological model and error model) could be considered a form of parameter updating, where the parameters are updated every year based on the most recent data. This concept of parameter updating, and the rolling approach used, will be made more explicit in the introduction.

As suggested, the Discussion section will be expanded to contrast the findings in this work against the relevant references cited (e.g. Luo et al. 2011, Brigode et al. 2013), to highlight where there are similarities or different findings.

- Concerning the data assimilation and model updating issue, the bibliography is poorer (see detailed comment for page 3). Many references could be added. Since the authors chose to use the GR4J model and since the state updating they chose is the same one as the approach adopted for the GRP model ('adaptation' of the GR4J model for forecasting, used by the French flood forecasting centres), it is also worth mentioning this work (see detailed comments below). Beyond the references issue, it may (should ?) be noted that this study explores the benefits of model updating for seasonal forecasting, whereas many, if not most, studies consider shorter lead-times. This aspect has to be mentioned, since it is well known that the effects of model updating most often vanish whe the lead-time increases. In my opinion, keeping benefits at large lead-time is one of the (surprising) key result of this study and may be usefully emphasized.

Thank you for the very constructive suggestions for relevant literature here and in the detailed comments. The literature review will be updated to include these references and points to improve the representation of the previous literature in the field of data assimilation.

See the response to detailed comments on emphasizing one of the key results of this work, that model updating still improved forecast performance at the lead-time of one month. This includes highlighting that this has not been focused on in the literature in the introduction section, and highlighting this in the abstract and conclusions. This change is proposed in the introduction:

The impact of assimilating observed data into CRR model state variables at the start of a probabilistic forecast on  has focused on short lead-times, however any benefits at longer lead-times (e.g. seasonal) has had limited evaluation.

A few methodological choices may deserve a little more discussion or explanation: The calibration algorithm is a rather complex one, but used with assumptions which are known to be not met in most cases (page 9, line 7: independent, homoscedastic residuals). Moreover, these assumptions are not consistent with the choice of the model error post-processor (a Box-Cox transformation is used in order to take into account the heteroscedasticity of these same residuals). Why did the authors pick a complex approach with unverified and inconsistent assumptions rather than a simpler one? It let the reader think that the authors used "components" available on the shelf or a pre-existing tool, which is quite understandable. But then they have to justify these choices (and why a so complex calibration method when much simpler ones are easily available?).

It was considered that a calibration method capable of estimating a posterior distribution of parameter values was required. This allowed the change in the distribution of suitable parameter values to be considered over time (Figure 8). In contrast, a simpler algorithm (gradient or evolutionary method) that provides a point estimate of the calibrated parameter values would not show the parameter identifiability (spread in values), and in turn the trends over time may not have been able to be separated from the variability in estimates. As such, the choice of algorithm is not considered overly complex, and a significantly simpler approach to estimating parameter uncertainty is not known to the authors.

The assumptions of the GR4J model likelihood function and the error model likelihood function are different, as they are applied in different situations:

- The GR4J model was calibrated at the daily scale to observed rainfall. The likelihood function used for the calibration of the GR4J model is arguably the simplest available, the standard least squares. While this approach is simple, it is expected to enable the model to capture high flows, as outlined. While the assumptions made by this likelihood may not met, the function is considered fit for purpose. This discussion in Section 3.2 will be expanded on to make this point clearer.
- The error model is applied based on the outputs from the GR4J model aggregated to the monthly time scale. The Box Cox transformation is used to capture heteroscedasticity and skew in residuals, which allows more reliable forecasts. Both observed and forecast rainfall were used to drive the GR4J model, and the error structures based on the forecast rainfall are very different from observed rainfall. As such, the errors, and error assumptions, would be expected to be different in these two cases.

Given the different objectives of the two models, and the different time scales adopted, this inconsistency is considered to be reasonable, though we agree in general the effects of this type of inconsistency warrant a separate investigation. This point will be included in Section 3.5. The approach used is not without precedent, as other forecasting systems implemented in Australia also adopt different objective functions at different stages of the modelling chain (Lerat et al., 2015).

Furthermore, one point is not discussed but may deserves some attention. Like the hydrological model, the model error post-processor is calibrated (not in a joint manner however). Why does the study on the impact of the calibration data period length on the calibration only focus on hydrological parameters and not on the post-processor parameters as well (mu, sigma)? This can indeed be treated independently (therefore not necessary in this article), but this research issue may be usefully mentioned. Are the post-processor parameters concerned by the rolling calibration?

The error model parameters are also determined using the rolling calibration approach, and this will be clarified in Section 3.5. This way any trends in the error model parameters are captured in the same way as those for the hydrological model. For the sake of brevity, a figure similar to Figure 8 has not been included for the error model parameters.

One element may also be better detailed: the results are given at a monthly scale (time step), whereas the GR4J model is a daily one: the way the GR4J model is run has to be precised. This is important, since the model is updated and effects of model updating decrease when the lead-time increases. However, it often does not only depend on the lead-time 'absolute' value but also on the number of time steps to reach this lead-time.

Good point. Details of the time aggregation of the GR4J model results and warm-up period will be added to the model description in Section 3.1 and 3.4.

In a nutshell, this article brings some interesting results, even in a field explored by many previous studies, and deserves publication. The suggestions made in order to improve the manuscript lead me to propose a moderate to major revision (however another round of submission afterwards does not seem necessary).

Thank you for the constructive comments that will help to improve the manuscript substantially.

**DETAILED COMMENTS**

- Page 2

Line 22 ("As these models are conceptual, they require calibration [...]"): they are not the only models that do so. Even the (so-called) physically-based models which could theoretically not need calibration, are most often calibrated, for various practical reasons.

Agreed, the manuscript will be updated to reflect that in fact most if not all environmental models require some level of calibration.

Lines 22 - 23: Brigode et al. (2013) show that this general a priori (longer calibration periods produce more robust parameters estimates) is not always verified. Therefore if this article is quoted (and it should be, in my opinion), it would be fair to indicate their results.

Agreed, and explanation of results of Brigode et al. (2013) will be included in the manuscript.

Lines 30 - 31: the definition of catchment non-stationarity which is proposed, is very interesting since it 'focuses' (restrains to) the physical object non-stationarity. However, it seems to be a binary state: the catchment is or is not stationary. Have the authors considered the notion of a "degree" of non-stationarity? Indeed, all the listed factors of non-stationarity are not expected to have the same consequences over the catchment behaviour. Might the rolling calibration approach be a tool to assess the relationship between "degrees" of non-stationarity and parameters evolution? (see also detailed comment on Fig. 3)

This is a good point, indeed natural systems are likely to be in varying degrees of non-stationarity, as nothing in nature is entirely static. This point will be included in the manuscript.

- Page 3

Line 1: is groundwater depletion a physical change (of the catchment) or the consequences of some of the listed catchment changes?

This clarification will be included, land use change has been shown to be a contributing factor to the observed groundwater depletion (Avery and Harvey, 2014, Brookes et al. 2017).

Lines 15 - 20: the bibliography review is rather poor: it gives some "extreme approaches" between the (too) simple GLUE and the very detailed BATEA (or similar approaches). Furthermore, GLUE is a quite old approach, giving a reference of 2008 is a bit strange (unfair?), as it appears more recent than much more advanced and sophisticated approaches, as those developed by Kavetski, Vrugt and others. Since the chosen approach is a model error post-processor, I suggest Krzysztofowicz and Maranzano (2004).

Thank you for the reference, it will be included. The reference for GLUE will be changed to Beven and Binley (1992); the reference to Krzysztofowicz and Maranzano (2004) will also be included.

Line 17: "using a model error post-processor" rather than "using a post-processor error model" ?

The terminology will be clarified. "post-processor error model" is preferred by the authors, as the error model is developed and applied after the hydrological model has been calibrated, rather than calibrated in conjunction with the hydrological model (i.e. post-processor), and "error model" emphasises that an error model has been used.

- Page 4

Line 3 ("up to one month"): I understood this paragraph as a bibliography review giving general results (not specific to some catchments). However, it gives some values of the "influence duration" of the initial state, which strongly depends on the catchment characteristics. I am pretty confident in the fact that it easy to find catchments where the impact of the initial conditions is important during several months (even years).

This paragraph is indeed intended to review general results, and the "up to one month" statement is from the cited references (Li et al., 2009; Wang et al., 2011). However, it is agreed that this is not a universal rule, and will change depending on the catchment. The sentence will be changed to:

The impact of initial catchment condition is particularly pronounced when forecasting over short lead times, *typically* up to one month (Li et al., 2009; Wang et al., 2011), *however this time frame is catchment dependent.*

Line 5: "warm-up" rather than "warmup"?

The change will be included throughout.

Lines 14-16: it may be specified that this impact has been deeply evaluated for shorter lead-times (this emphasizes the character of novelty of the study). Furthermore, I disagree with the second sentence as it has been shown that the impact decreases quite fast (for

many not too slow catchments) and is almost negligible at a seasonal scale (see e.g. Berthet et al. 2009 that the authors quote elsewhere). That is one very interesting aspect of the results of the submitted study.

This was the intent of this paragraph, to highlight that the impact at seasonal scales has not had extensive evaluation. The paragraph will be changed to:

The impact of assimilating observed data into CRR model state variables at the start of a  probabilistic  forecast has focused on short lead-times, however any benefits at longer lead-times (e.g. seasonal) has had limited evaluation.

Line 18: I suggest to precise "calibration periods choice" or "calibration periods length" rather than only "calibration periods".

The change will be included throughout.

Line 21: to enhance "seasonal" forecasting skill?

Yes, this will be added.

Line 22: Does the article "demonstrate" that calibration period choice can affect forecast skill (that is quite known) or does it assess how much it does so?

The aim will be updated to focus on the assessment, rather than demonstrating a known influence.

- Page 5

Line 11: is it the gauge "A2390514" rather than "A21390514"?

Correct, the typo will be corrected.

Lines 26 - 27: an hydrograph may be useful to support this information.

A hydrograph of flow in Drain M will be included to demonstrate the variability.

- Page 6

Lines 3-13: is the description of the model developed by eWater Source useful for the reader. If I understood correctly, it is not directly related to the model used in this study. If so, this might confuse a bit the reader. E.g., I am not sure that the assumption of a constant inflow of salinity (which is not discussed) is needed by the reader to understand how the authors worked to answer to the scientific questions (which are the core of the article). The multi-objective nature of the calibration is also of no use for the rest of the study. If the fact that this model is used in practice had consequences on the methodological choices for this study, then the authors may consider explaining it (and discuss results with respect to it and to the specific context of wetland management).

As pointed out earlier, we agree that Section 2 includes superfluous detail that will be removed. This section on the separate water balance model in eWater Source will be either substantially shorted or removed altogether.

Line 14 ("To use this model for to inform operations"): it is always tricky for a non native English speaker to ask so to native ones, but may the authors check English here?

Correct, the typo will be corrected.

Line 15: "lead-time" rather than "leadtime"?

Correct, the typo will be corrected throughout.

Line 15: to fully understand the implication of the choice of the 1-month lead-time, it is necessary to know that the CRR model is a daily one (not only because the model is updated). However this information is given at subsection 3.1 (and not very explicitly: the reader has to know that GR4J is a daily model)

Good point, this information will be included as part of the aims and objectives in section 1.3.

Line 19 ("reasonable forecast skill is expected to be possible compared to longer forecast horizons"): may the authors provide some references? How much are the performances expected to decrease for longer lead-times? Furthermore, why did the authors choose to focus on a single lead-time? The evolution of the benefits of the model updating, with respect to the lead-time, in a context of seasonal forecast, would be a very interesting result.

We agree that this comparison of longer lead times, and how they influence the state updating performance, would be a very interesting study. However, it was considered beyond the scope of this work. The statement is based on the fairly obvious observation that the skill of rainfall forecasts is expected to reduce the longer the forecast horizon, and as such the skill of the streamflow forecasts will also reduce. The sentence will be changed to:

and 3) the skill of rainfall, and hence streamflow, forecasts is expected to decrease as the lead-time increases.

Line 20 ("The mean annual rainfall for the region is in the range 600-675 mm"): page 5, lines 3 and 4 suggest some spatial variability. How strong is it? (600 to 675 mm is not very strong difference, compared to some other climates around the globe).

As pointed out, ~10% range is not very strong, but it is spatially consistent, as opposed to representing annual variability. The manuscript will be updated to clarify that there is a rainfall gradient from south to north.

Line 20: is it useful to precise what "FAO56" stands for?

The acronym will be expanded and a reference included (Allen et al., 1998). It stands for Food and Agriculture Organization of the United Nations, and paper 56 relates to guidelines for computing crop water evapotranspiration.

- Page 7

Lines 3 - 4 ("2 rainfall hindcasts [...] were downscaled to the single rainfall gauge scale"): just to be sure, does it mean to the pixel where the gauge is?

This will be expanded on. The pixel size is ~250 km, which tends to smooth out the rainfall events. The downscaling process, mapping the pixel where the rainfall gauge is to the gauge data, is used to restore more representative rainfall events.

Lines 15 - 25: the authors may consider whether this paragraph would not be better written earlier (e.g. among the first paragraphs of section 2).

Agreed, the paragraph on the catchments and where they flow will be moved to the start of section 2.

Line 20 ("It should also be noted that releases from Bool Lagoon [...]"): why is it important to understand this scientific study? (I worry about missing something useful for the interpretation of the results)

This will be clarified. The point being made was it is rare that the upstream catchment contributes to the downstream catchment, as releases from Bool Lagoon are rare.

Lines 30 and following: are the details about the streamflow measurements devices useful?

These details help establish that any perceived non-stationary trends are unlikely to be due to streamflow instrumental measurements. We will clarify this issue and shorten this section to avoid superfluous details.

- Page 8

Line 3: I agree with the fact that indicating the data are of good quality and too often not done, but if the authors want to demonstrate the quality of the rating curves, they may add some information about the number of years during which the 78 and 166 gaugings have been achieved and how much often the rating curves have been modified.

It is agreed that this is an important component of streamflow data quality. Information on rating curve modifications will be included in the reworking of this section (see previous comment).

Lines 9 - 20: since catchment non-stationarity is an important issue for this study, I suggest to make this paragraph a subsection dedicated to this topic (here).

This is a good suggestion which will be adopted.

Lines 23 - 24 ("GR4J [...] explicitly accounts for non-conservative (or 'leaky') catchments"): I agree. However, it should be kept in mind that GR4J has not been designed nor is known to achieve good performances for karstified catchments (mentioned page 7, line 13). Moreover, I am not convinced it is quite appropriate for ephemeral catchments (as suggested by line 21, page 11).

It is agreed that GR4J may not be ideal in ephemeral catchments, as the exponential decay relationship used in the storage reservoirs cannot completely dry out. However, this is a relatively theoretical consideration, for example, if any simulated flow below that which could be adequately measured is considered to be zero, ephemeral behaviour can be represented. As outlined, previous studies have demonstrated good performance for Australian conditions, including ephemeral catchments (Coron et al., 2012; Guo et al., 2017). Westra et al. (2014) will also be added to this list, who applied GR4J in a similar location in southern Australia. Considering alternate model structures was beyond the scope of the study, however, it is possible (likely?) that more appropriate model structures could be identified in future work.

Lines 28 - 31: may the authors explain what motivates their choice of adding a 5th free parameter to calibration? Is it important for their particular catchments or for their methodology in this study?

As noted in the previous comment, GR4J was not designed for this application. The catchments considered have a relatively slow response, and it was considered that the pre-specified split to the routing store of 0.9 may be too low for these catchments. As can be seen in Figure 8, this turned out to be the case for the calibrated parameter values, where higher values of the split parameter were found, in particular for C2. This point will be clarified in the manuscript.

- Page 9

Lines 10 - 12 ("this function [RMSE] provides a focus on the highest flow in the time series, where the majority of the runoff occurs"): it is not necessary. It provides a focus on the largest absolute errors, which indeed most often occur for the largest flows. However, consider a hypothetical model whose errors would be only on low flows.

We agree, the original statement was not strictly correct. The statement will be updated accordingly.

Line 26 ("External influences include model structural limitations [...]"): this confused me, after reading the catchment non-stationarity given on pages 2 & 3. Does it suggest that parameters variation due to structural deficiencies would be considered here?

We agree this sentence does not add value to the discussion in Section 3.3. It will be removed to avoid confusion.

- Page 10

Lines 3-5: Would not it be useful to emphasize the trade-off between a longer calibration period to reduce the parameter uncertainty and a shorter calibration period to mainly take into account the most recent dynamics in the introduction section?

This tradeoff is indeed a key aspects of the study; we agree with the reviewer that it warrants stronger emphasis in the introduction.

Line 13-14: the literature review is also poor about data assimilation and model updating. Generic references may be Refsgaard (1997) and Liu and Gupta (2007). Since the chosen updating approach is the same as the one used for the GRP model (which is a mere adaptation of GR4J for forecasting purposes), I suggest to refer to the work of the team which developed these models. The authors may pick Tangara (2005) and Berthet (2010), both in French, which described the numerous tests of different updating approaches made by the GR4J research team (some of them discussed in section 5! See comment below) and detailed the resulting GRP model. They may prefer Berthet et al. (2010), which provides a much shorter description of the model and the updating techniques but also discusses the impact of the largest errors on the RMSE-based criteria values (see discussion page 9). For a detailed description of the GRP model, the authors may also consult: https://webgr.irstea.fr/en/modeles/modelede-prevision-grp/fonctionnement-grp/. Moreover, since sequential approaches such as ensemble Kalman filter and particle filters are mentioned, I suggest also to add references to Moradkhani et al. (2005, 5005b) and Weerts and El Serafy (2006).

Thank you for the very constructive suggestions. The literature review will be updated to include these references and points.

Lines 17 - 27: a flowchart would greatly help the reader.

It is considered that the equations provided are the clearest approach to explain the exactly methodology used to update the routing store to simulate the observed flow. Further details on this approach are provided in the original work, Demirel et al. (2013). This link to the earlier work will be made clearer in the revised manuscript.

Line 27: "where X3 is the estimated runoff model parameter" rather than "where X3 is an estimated runoff model parameter"?

This change will be made.

- Page 11

Lines 3 - 4 ("particularly when used to update both model state variables and model parameters"): I agree with the authors, but is it relevant here? (since parameters are not updated here).

It is agreed that this is out of place. This paragraph will be removed and integrated with the revision of the literature review, along with the comment on page 10 Line 13-14.

Line 12 ("Depending on the case"): this is not clear, until the reader reaches section 3.7.

This sentence will be reworded as follows:

As outlined in Section 2.1, both observed and forecast rainfall has been considered. When observed rainfall was used, the predictions used were those from the daily hydrological model, aggregated to the monthly time step. When the ensemble of forecast rainfall was considered, the hydrological model predictions were the median across the ensemble of the hydrological model predictions each day, and then aggregated to the monthly time step.

- Page 12

Line 5: why do the authors prefer to sample the (normalized) residuals rather than picking a number of calculated quantiles (from the Gaussian distribution)?

A direct selection of quantiles would provide an analytical approach to represent the distribution. However, this can be difficult, as the calculated quantiles in normalised space do not correspond to the same quantiles in un-normalised space, due to the combination of the transformation, the truncation of very low flows and the parameter uncertainty. The Monte Carlo Simulation approach used is more computationally intensive, but the most robust. This approach is also more generalised, and would be required is more complex error models were used. For these reasons the approach used has been retained.

Line 18: may the authors explain the choice of the 0.05 and 0.95 as normalized extrema values?

This range was adopted for data visualisation purposes only. If the $0 - 1$ range were used, the worst-performing case for each metric would correspond to zero area in Figure 4 and would not appear in the plot area. We decided to avoid this to avoid the impression that some information is missing.

- Page 13

Lines 17 - 21: as pointed out by the authors, the reference distribution has an 'unfair' advantage. Then may the authors explain this choice? Why have they not chosen a simple naive forecast model?

The reference distribution of the monthly streamflow is considered a simple naïve forecast model. Other approaches could be used, autocorrelation with last month's streamflow, for example. The unfair advantage referred to is that the reference distribution has been

calculated using all data, which can be from the "future" for earlier forecast periods. However, it is expected that the forecast models tested should be able to perform better than an uninformed climatology, and as such is considered a useful baseline for the calculation of forecast skill. The paragraph will be changed to:

The reference distribution for each month is calculated as the empirical distribution of all observed data in that month. Note that all of the observed data are used to estimate the reference distribution (including data in the forecast verification period), however, this is still considered a suitable simple naïve forecast model that is useful as a baseline for the assessment of forecast skill. CRPS$_{SS}$ values less than zero indicate the reference distribution produces better performance than the forecast model.

Line 23: check the formula. There is missing sum for the denominator.

This typo will be corrected.

- Page 14

Line 6: is the rainfall forecast used here the ensemble forecasts described in subsection 2.1? I don't think it obvious. If not, why were the ensemble described?

Yes, the ensemble forecasts were used in all cases. This will be clarified here.

Line 6: I found only 2^4 = 16 cases (model with or without updating; 2 calibration period lengths; 2 catchments and 2 rainfall forcings). What do I miss?

Well-spotted - this is an error and will be corrected to 16.

- Page 15

Line 6 ("Any detrimental impacts"): check English.

The sentence will be reworded as follows:

The state updating could be seen to reduce the accuracy of the forecasts occasionally when the model storage was updated to represent a zero flow, and then a low flow that did occur in the following month was underestimated, e.g. in January 2002 for C2 in **Error! Reference source not found.**.

Line 8 (and followings): I suggest to precise "calibration period length" rather than only "calibration period"

This will be corrected throughout.

Line 16: the differences are not much smaller for catchment C1. How much are they significant?

The original statement in the manuscript is correct, the difference between values was smaller for every metric and case when state updating was used. The anomaly is the precision metric for C1 with observed rainfall, which increased for the longer calibration period where all the others decreased. Nonetheless, the differences were still smaller for the "with state updating" case compared to "without". Significance tests have not been calculated due to the lack of replicates. No change is proposed.

Line 19 ("the differences were more pronounced for the most practically relevant cases with forecast rainfall [...]"): this is of particular interest for operational purposes (e.g., forecasts) and is worth being emphasized.

This is a good suggestion and will be emphasized in the revision.

Line 20: I don't understand how the model error post-processor compensates the introduced errors. I thought that it only assesses them.

The normalised residual can have a non-zero mean, which can compensate for biases in the hydrological model predictions. But the main purpose of the error model is to quantify the uncertainty in forecasts, not to compensate for errors. For clarity, the paragraph will be changed to remove this reference:

The differences were more pronounced in the more practically relevant case with forecast rainfall, which introduced further errors to be compensated by the state updating approach, and the postprocessor error model.

Lines 23-24 ("catchment C1 had been identified to have a substantial reduction in the rainfall-runoff relationship over time"). As discussed below (comments on Fig. 3), the catchment appears as rather stationary up to 1990 (approximately) and then also more or less stationary from 1990 to 2010. If it is so, how can the difference in calibrated parameters obtained with the 2 different calibration period lengths for years 2009 2010 be explained by this change around 1990?

The 20-year calibration period was 1989-2008. The 10-year calibration period was 1999-2008. As such, the early 1990s are part of the 20 year calibration period, but not of the 10 year calibration period. The results indicate that the 20 year calibration period resulted in model parameters that produce more flow the 2009 (and 2010) validation year than the 10 year calibration period. This suggests that streamflow data from the 1990s are substantially different from streamflow data from the 2000s. The manuscript will be changed as follows:

From **Error! Reference source not found.**, it can be seen that the model calibrated to the longer period *(e.g. 1989-2008 for the 2009 forecast period)* overestimated the observed flow in 2009 and 2010 in catchment C1, whereas the model calibrated to the shorter period *(e.g. 1999-2008 for the 2009 forecast period)* provided a better reflection of the catchment response.

- Page 16

Line 16 ("A model fitting anomaly resulting from a shorter calibration period"): did the author investigate this "anomaly"? How can it be explained?

This was not investigated further. It is assumed to be due to reduced parameter identifiability. A different combination of parameters (e.g. higher X4 and lower X2 and *split* compared to the periods before and after) resulted in similar values for the objective function for this particular period. This discussion will be added to the manuscript.

Line 26: this is interesting at a seasonal scale, since it has been shown that hydrological models are "stable", i.e. the updating effect vanishes after a number of time steps (e.g. Berthet et al. 2009 at a hourly time step: then after a few days at most for a large majority of the tested watersheds).

Thank you.

Line 29 ("As the range in model predictions should be reduced by forcing the model to simulate the observed streamflow at the start of the forecast period"). Is there any confusion between precision and sharpness? Model updating increase sharpness and precision (at least for the shortest lead-times).

The term "precision" was used in this work as a synonym for "sharpness", as outlined on Page 12 line 24. However, as "sharpness" is a term more generally used in the foreasting community, "precision" will be changed to "sharpness" throughout.

Line 30 ("the trade-off for an increase in precision would typically be a reduction in the reliability of the predictive uncertainty"): again I assumed that the authors meant "sharpness". I suggest to write that this trade-off for an increase in sharpness \*\* may \*\* result in the reduction of the reliability, if the authors do not provide a general (theoretical) explanation. As much as I know, this is a common feature, but exceptions should exist, and the last sentence of the paragraph (page 17, lines 1 - 2) says so.

See above, "precision" will be changed to "sharpness" throughout. Nonetheless, this is a good point, as the result cannot be proven to be generic. The qualification "may" will be added to this sentence.

- Page 17

Line 4 ("update the GR4J production store along with the routing store"): this has been tested by the GR4J team (Berthet, 2010), with no significant improvement in a forecasting context at a hourly time step.

The manuscript will be updated to refer to this study and highlight the potentially limited benefit.

Lines 4 - 5 ("This could be expected"): may the authors give some explanation to found this idea?

The manuscript will be modified to outline this hypothesis. At some times, the routing store was updated to be empty, but the model still overstated the observed flow. Also reducing the size of the production store could further reduce the simulated flow, to match the observed flow in these cases. It should be noted this approach has not been tested to date.

Lines 3 - 9: this paragraph is very interesting, but how is it related to the scientific issues developed in this article?

This paragraph was intended to point toward further work to improve the approaches used. It will be reworded to make this clearer.

Lines 16 - 19: in my opinion, this is the (or one of the) key findings. How may the authors emphasize it, rather than putting it at the very end of the article?

Agreed. This finding will be added to the abstract and conclusions, and the second aim of the study will be framed to emphasize this aspect of the investigation.

Line 30 ("in most cases"): the study was driven only on 2 catchments... It should be pointed out that a work over a (much) larger number of catchments is needed to ensure the generality of these interesting results.

We agree - our intent was to say that this finding held in most cases considered in the study, i.e. the different metrics with and without state updating. The sentence will be reworded to avoid confusion.

- Page 19

Tab. 1: the authors may usefully add the parameters meanings and their units (and a GR4J flowchart aside).

Good point - this information will be added to the manuscript.

- Page 20

Fig. 1: the drain M is not given in the legend and has the same color as catchment boundaries, which makes it difficult to identify.

Good point - the map will be adjusted accordingly.

Fig. 2: what are the upper and lower bounds? Minimum and maximum of the ensemble? Some predictive quantiles such as 0.05 and 0.95? If the latter, is there any information about the reliability?

That is correct, the upper and lower bounds on the figure legend refer to the minimum and maximum of the ensemble. This will be clarified in the figure caption.

- Page 21 (Fig. 3)

I wonder if there is not a "sudden" change around 1990: catchments C1 and C2 look quite stationary up to 1990, and also after 1990. If it is so, can it really be explained by plantation forestry expansion (which is more a "continuous" factor of non-stationarity). Furthermore, can this question the relevancy of the rolling calibration which might be better adapted for smooth non-stationarity?

A step change is arguably more likely to have taken place over the mid-late 1990s. The early 1990s are also quite wet, which may result in steeper slopes for these years. A cause for a sudden change in the streamflow response around 1990 is not known to the authors. Reductions in groundwater levels due to the forestry expansion has been attributed in other studies (Avery and Harvey, 2014, Brookes et al. 2017).

In any case, identifying such a distinct step change, or even a continuous change, is very difficult to do with certainty. The difficulty in identifying such step changes, and the idea of using the rolling calibration approach as a means to overcome this, will be added to Section 3.3. The discussion on Page 17, lines 16-19, will also be amended to more clearly address this point.

- Page 22 (Fig. 4)

It is very important to insist on the fact that the plot gives relative values of the metrics (higher is better) which are then not consistent with the formulas and details given in subsection 3.6. I was first confused becaused I did not notice (at first) the mention in the y-label (even if I admit that it is written in (sufficiently) large characters). I suggest to change the criteria described in section 3.6 to give only their relative values.

The criteria description in Section 3.6 will be updated to match the values used in Figure 4, and this information will also be added to the caption.

- Page 23 (Fig. 5)

Why are the results plotted only up to 2005?

This was done for illustrative purposes only. Some of the desirable detail in the hydrographs is difficult to see when the whole time series is plotted (~20 years). As such, a subset of the data was shown, to provide a trade-off between showing all of the results, and making a clear point. This point will be stated in the paper, and the full plot provided as supplementary material.

Page 25 (Fig. 7)

Why are the results plotted only from 2008 to 2010?

This was done for illustrative purposes only – same reason as in the point above. Again, this point will be stated in the paper, and the full plot provided as supplementary material.

REFERENCES

Allen, R.G., Pereira, L.S., Raes, D. and Smith M. (1998). Crop evapotranspiration - Guidelines for computing crop water requirements. *FAO Irrigation and drainage Paper 56*. Food and Agriculture Organization of the United Nations, p.300.

Avery, S. and Harvey, D. (2014) How water scientists and lawyers can work together: A 'down under' solution to a water resource management problem. Journal of Water Law 24(2):45-61

Brookes J.D., Aldridge, K., Dalby, P., Oemcke, D., Cooling, M., Daniel, T., Deane, D., Johnson, A., Harding, C., Gibbs, M.S., Ganf, G., Simonic, M. and Wood, C. (2017). Integrated science informs forest and water allocation policies in the South East of Australia. Inland Waters *(In Press).*

Lerat, J., Pickett-Heaps, C.A., Shin, D., Zhou, S., Feikema, P., Khan, U., Laugesen, R., Tuteja, N.K., Kuczera, G., Thyer, M., and Kavetski, D. (2015) Dynamic streamflow forecasts within an uncertainty framework for 100 catchments in Australia [online]. In: 36th Hydrology and Water Resources Symposium: The art and science of water. Barton, ACT: Engineers Australia, 2015: 1396-1403.

---

## Referee Comment (RC3) · Anonymous Referee #3 · 31 Aug 2017

This manuscript presents an original study focused on two important aspect of monthly streamflow forecasting: state updating and the selection of an appropriate calibration period. The watershed on which the methods are implemented and tested is a very interesting case study. This semi-arid watershed is extensively impacted by human intervention, as it includes diversions and "Drain M", so runoff water can be directed either toward the north to benefit wetlands or toward the south to maintain fish ecosystems and comply with other constraints. Adequate management of this watershed appears very difficult and it is evident from the authors' description of the situation that

monthly hydrological forecasts are essential. The authors show that selecting a calibration period during which the conditions are close to the expected forecast conditions can improve forecasts performance substantially.

In my opinion, this is a well-written paper (very clear!) that brings interesting novel knowledge in the field of ensemble monthly streamflow forecasting. I also find it completely appropriate for publication in HESS, especially since the case study raises issues regarding water management and tradeoffs between ecosystem services and human activities.

I appreciate that the authors included a short discussion about the uncertainty related to the gauging measurements (page 8 first paragraph). I also like the idea of adding the "split" parameter to GR4J in the calibration process.

The conclusions drawn by the authors regarding the tradeoff between the length of the calibration database versus its "representativeness" is interesting. I appreciate that they recognize that said conclusions might be limited to their specific case study and that further investigation for a wider range of hydro-climatic regimes would be needed.

I only have very few minor comments and suggestions that I think could improve the paper. I strongly recommend that it be published in HESS.

*Minor comments*:

1. Why did you consider only the median of the hydrological model predictions rather than the whole ensemble?

This question kept bugging me all along while I was reading. You have access to meteorological ensemble forecasts (page 6 line 30 to page 7 line 5). They are expected to account for the uncertainty related to the meteorological conditions (or at least a part of this uncertainty). Why then did you not keep all the scenarios after passing everything through the hydrological model? Is it to make it more comparable to the case where the hydrological model is forced by (deterministic) observations? If so, I

personally don't see why this would be necessary. And then after, you use a statistical method to dress the median back to an ensemble. Is it because you found out that hydrological ensemble forecasts built only from meteorological ensembles were under-dispersed and would have needed post-processing?

In my opinion, those choices (i.e. using the median, thus ignoring the other ensemble members, and then dressing the median into an ensemble) really need further explanations/justifications.

2. There are a couple of (very minor) elements that could be clearer

- Page 5 line 30: please add a reference for the Ramsar list. I didn't know this list before reading the manuscript so I looked it up on the web. I think that a reference would be helpful to be sure that other readers like me know what you are talking about.

- Page 9 line 21: What is a "burn-in" period? At first I thought it was a synonym of "warm-up period" in the context of the DREAM algorithm, but I am really not sure.

3. There are a few typos in the manuscript and I am unsure of the spelling for 1-2 words:

- Page 3 line 27-28: parenthesis typo, replace "(McInerney et al, 2017)" by "McInerney et al. (2017)"

- Page 11 equation (5): Something seems wrong with the curly brace

- Page 11 line 1: I think that the sentence "(. . .) are available, for example, the ensemble Kalman filter (. . .)" should be split, as in: "(. . .) are available. For example, the ensemble Kalman filter (. . .)"

- Page 11 line 12: day "n" and month "t" should be in italics.

- Page 12 line 17: I think that "straightforward" should be written in one word.

- Page 13 equation 12: A summation seems to be missing at the denominator. Also, I

don't think there should be a "t" index for the average streamflow, since by definition it is independent from time (it is the average of the time series).

- Page 14 line 6 (and several other places in the manuscript): Why do you write "observ*ed* rainfall" but not "forecast*ed* rainfall" I am not a native English speaker so perhaps I am completely wrong, but I could not help but finding this strange.

- Page 14 line 16: Is there a "to" missing in "The changes due adopting (...)"?

---

## Author Comment (AC2) · 18 Sep 2017

This manuscript presents an original study focused on two important aspect of monthly streamflow forecasting: state updating and the selection of an appropriate calibration period. The watershed on which the methods are implemented and tested is a very interesting case study. This semi-arid watershed is extensively impacted by human intervention, as it includes diversions and "Drain M", so runoff water can be directed either toward the north to benefit wetlands or toward the south to maintain fish ecosystems and comply with other constraints. Adequate management of this watershed appears very difficult and it is evident from the authors' description of the situation that monthly hydrological forecasts are essential. The authors show that selecting a calibration period during which the conditions are close to the expected forecast conditions can improve forecasts performance substantially.

In my opinion, this is a well-written paper (very clear!) that brings interesting novel knowledge in the field of ensemble monthly streamflow forecasting. I also find it completely appropriate for publication in HESS, especially since the case study raises issues regarding water management and tradeoffs between ecosystem services and human activities.

I appreciate that the authors included a short discussion about the uncertainty related to the gauging measurements (page 8 first paragraph). I also like the idea of adding the "split" parameter to GR4J in the calibration process.

The conclusions drawn by the authors regarding the tradeoff between the length of the calibration database versus its "representativeness" is interesting. I appreciate that they recognize that said conclusions might be limited to their specific case study and that further investigation for a wider range of hydro-climatic regimes would be needed. I only have very few minor comments and suggestions that I think could improve the paper. I strongly recommend that it be published in HESS.

Thank you for the constructive comments that will improve the manuscript.

*Minor comments*:

1. Why did you consider only the median of the hydrological model predictions rather than the whole ensemble?

This question kept bugging me all along while I was reading. You have access to meteorological ensemble forecasts (page 6 line 30 to page 7 line 5). They are expected to account for the uncertainty related to the meteorological conditions (or at least a part of this uncertainty). Why then did you not keep all the scenarios after passing everything through the hydrological model? Is it to make it more comparable to the case where the hydrological model is forced by (deterministic) observations? If so, I personally don't see why this would be necessary. And then after, you use a statistical method to dress the median back to an ensemble. Is it because you found out that hydrological ensemble forecasts built only from meteorological ensembles were underdispersed and would have needed post-processing?

In my opinion, those choices (i.e. using the median, thus ignoring the other ensemble members, and then dressing the median into an ensemble) really need further explanations/justifications.

The focus of this work was to investigate the impact of the state updating and calibration periods on streamflow predictive uncertainty, and a current best practice post-processing approach was adopted to estimate this predictive uncertainty. It is common practice to use deterministic inputs, often based on summarising an ensemble of forecasts using summary metrics (e.g. mean or median) as the input to post-processing methods (e.g. Lerat et al., 2015; Matte et al., 2017; Schepen et al., 2017; Wani et al., 2017). As such, is was considered beyond the scope of this work to also consider improving methods for applying post-processing models.

However, it is acknowledged that that by only considering the median streamflow simulated across the forecast rainfall ensemble that some information is lost. The paper will be modified to include the references above to outline that the current practice for applying post-processor error models has been adopted in Section 3.5, and highlight that further work is needed to more fully utilise the information from ensemble forecasts when developing post processing models.

2. There are a couple of (very minor) elements that could be clearer

-       Page 5 line 30: please add a reference for the Ramsar list. I didn't know this list before reading the manuscript so I looked it up on the web. I think that a reference would be helpful to be sure that other readers like me know what you are talking about.

The reference Matthews (1993) will be added.

-       Page 9 line 21: What is a "burn-in" period? At first I thought it was a synonym of "warm-up" period in the context of the DREAM algorithm, but I am really not sure.

Effectively, the two approaches are the same. The term "warm-up" typically used in hydrological modelling, and "burn-in" typically used in Markov Chain Monte Carlo modelling. The commonly used terminology is proposed to be maintained, with a qualifier added in the DREAM description for clarification:

A maximum of 25,000 evaluations were used, including a burn-in period *(similar to the warm-up period used in hydrological modelling)* where the initial samples were discarded…

3. There are a few typos in the manuscript and I am unsure of the spelling for 1-2 words:

- Page 3 line 27-28: parenthesis typo, replace "(McInerney et al, 2017)" by "McInerney et al. (2017)"

The typo will be corrected.

- Page 11 equation (5): Something seems wrong with the curly brace

The curly brace is commonly used notation for conditional expressions, in this case to avoid dividing by 0 when $\lambda=0$. No change is proposed.

- Page 11 line 1: I think that the sentence "(…) are available, for example, the ensemble Kalman filter (…)" should be split, as in: "(…) are available. For example, the ensemble Kalman filter (…)"

It is agreed that the suggested change is clearer and will be adopted.

- Page 11 line 12: day "$n$" and month "$t$" should be in italics.

The change will be adopted.

- Page 12 line 17: I think that "straightforward" should be written in one word.

The change will be adopted.

- Page 13 equation 12: A summation seems to be missing at the denominator. Also, I don't think there should be a "$t$" index for the average streamflow, since by definition it is independent from time (it is the average of the time series).

The summation typo will be corrected. It is agreed that the index $t$ for the average is also incorrect. Thank you for picking these up.

- Page 14 line 6 (and several other places in the manuscript): Why do you write "observ**ed** rainfall" but not "forecast**ed** rainfall" I am not a native English speaker so perhaps I am completely wrong, but I could not help but finding this strange.

This is a strange nuance of the English language. Both "forecast" and "forecasted" can be used as past tense for the verb forecast. It was considered that "forecast" is more commonly used in the literature and is proposed to be maintained throughout.

- Page 14 line 16: Is there a "to" missing in "The changes due adopting (…)"?
The typo will be corrected.

**References**

Lerat, J., Pickett-Heaps, C., Shin, D., Zhou, S., Feikema, P., Khan, U., Laugesen, R., Tuteja, N., Kuczera, G. T., M, and Kavetski, D.: Dynamic streamflow forecasts within an uncertainty framework for 100 catchments in Australia, 36th Hydrology and Water Resources Symposium: The art and science of water, Barton, ACT, 1396-1403, 2015.

Matte, S., Boucher, M. A., Boucher, V., and Fortier Filion, T. C.: Moving beyond the cost–loss ratio: economic assessment of streamflow forecasts for a risk-averse decision maker, Hydrol. Earth Syst. Sci., 21, 2967-2986, 2017.

Matthews, G. V. T.: The Ramsar Convention on Wetlands: its History and Development, Ramsar Convention Bureau, Gland, Switzerland, 1993.

Schepen, A., Zhao, T., Wang, Q. J., and Robertson, D. E.: A new method for post-processing daily sub-seasonal to seasonal rainfall forecasts from GCMs and evaluation for 12 Australian catchments, Hydrol. Earth Syst. Sci. Discuss., 2017, 1-27, 2017.

Wani, O., Beckers, J. V. L., Weerts, A. H., and Solomatine, D. P.: Residual uncertainty estimation using instance-based learning with applications to hydrologic forecasting, Hydrol. Earth Syst. Sci., 21, 4021-4036, 2017.

---

## Author Response (AR1)

1) This article investigates 2 scientific issues in the context of rainfall-runoff seasonal forecasting: (a) the advantages of state(s) updating and (b) the sensitivity of the choice of the data used for calibration (calibration period length). These 2 scientific issues have been / are widely discussed in the hydrological community. Authors choose to put them in the context of complex wetland management application. This intention is relevant, since these issues are of crucial importance for operational matters.

First, it is worth noting that the manuscript is most often very clear (in particular, the introduction is efficient). Some suggestions are made below (detailed comments) to make the manuscript clearer (some parts are easier to understand when checked again after a further reading). The methodology is quite well detailed (I reckon that it is sufficient for anyone who wishes replicating the study) and the results are well presented.

However, many (too many ?) details concerning the application context are provided (section 2). I am afraid that I missed understanding how they infer with the scientific issues: results and discussion section do not make clear to me whether and how this particular context has implication on the way these issues are dealt with and on the results of the study. In a similar way, some details are given about the data used in an operational context, but it is not clear how they impact the results of this study. For example, this is the case of the precipitation forecasts (see detailed comments). Indeed all the results are not discussed in depth, with respect to these options (e.g., results obtained with observed rainfall versus results obtained with forecasted rainfall) and with respect to the context and practical purposes of the wetland management (whereas this appears in the submitted title): results are presented in only 2 pages and a half and the discussion is shorter (1 page). Even if all the tested cases are very useful for the specific case study and application, they are then not fundamental for the reader who focuses more on the 'generic' scientific issues than on the specific context of wetland management. The authors should consider removing them in order to make the reading and the analysis easier, rather than providing everything they learnt from their case study (again: even if it is quite interesting per se). They may prefer explaining how their findings are related to their specific case study and practical application.

One of the topics of interest for the sub-seasonal to seasonal hydrological forecasting special issue was user needs for seasonal forecasts. As such, more detail than typical on the case study application to wetland management was included in the manuscript. However, it is agreed that this is a distraction from the more generic scientific issues of interest to most readers. As such, section 2 has been shortened and more targeted. See responses to detailed comments below for further changes in this regard.

It is believed that all the test cases considered remain relevant to the contribution of the manuscript and general scientific issues, i.e. with and without state updating, two different calibration period lengths, and observed and forecast rainfall. The least contribution may come from the two rainfall sources, but these cases represent two different tests. The observed rainfall allow the effect of the changes on model performance to be isolated, without the effects and errors associated with the forecast rainfall propagating through. However, the forecast rainfall case is necessary for the practical application, and without this case the results cannot be considered realistic.

2) As mentioned previously, the 2 scientific issues have been explored by many previous studies. That is why this article has to do thorough review of literature in order to emphasize on the novelty of their study or to compare their results to those of other studies:

-        The way how non-stationarity is treated is very satisfying. The explicit distinction between physical catchment non stationarity and other model non-stationarity is necessary (while not always made); it is introduced in a very clear manner. I only suggest the authors to give a more explicit definition of the model parameters (the discussion is indeed implicitly present behind), since some previous studies proposed parameter variations to compensate many different non-stationarities (up to model structural deficiencies), as nicely pointed out in the introduction. This issue is particularly relevant in this study because the authors chose to update their model but only selected state updating, while many other approaches exist, one of them being parameter updating: this choice, which is very consistent, may be better explained. While not being a specialist of the choice of data calibration, I found the quoted references relevant. I only wish that these articles (e.g., Luo et al., 2011, which clearly inspired the methodology adopted by Gibbs et al.) would have been quoted not only in a generic way but also in sections 4 (Results) and 5 (Discussion) as benchmarks for the results: the results confirm previous studies in a large part; is there any interesting difference?

The description of the model parameters and structure has been expanded in Section 3.1 and Table 1. The rolling approach used to calibrate the model parameters (both the hydrological model and error model) could be considered a form of parameter updating, where the parameters are updated every year based on the most recent data, this has been clarified in Section 3.3. The introduction has also been updated to outline the approach to parameter updating based on analogous periods in the historical record:

*The degradation in model predictive performance due to catchment non-stationarity can impact on the decisions informed by these forecasts. To address this concern, a number of studies have calibrated model parameters to subsets of the available data, by attempting to find periods in the historical record that are analogous to conditions expected in the prediction time period, and by tailoring the time period selection to compensate for deficiencies in the model structure or input data (Brigode et al., 2013; de Vos et al., 2010; Luo et al., 2012; Vaze et al., 2010; Wu et al., 2013; Zhang et al., 2011). Often there is a trade-off between the benefits of a longer calibration period which exposes the model to a more diverse range of conditions and tends to improve parameter identifiability, versus the benefits of a*

*shorter calibration period that exposes the model to the most recent – and hence often the most relevant – dynamics in the catchment. Demonstrating and understanding the impact of this trade-off on model predictive performance is a key research gap pursued in this study.*

The Discussion has been updated to contrast to previous studies and the interesting differences found in this work. Most notably, this difference is the benefit of state updating at a lead time longer than a few days. This is outlined in Section 5.1:

*Most previous studies have used state updating in a short term flood forecasting context, and found limited effect of the initial conditions after a number of days (e.g. Berthet et al., 2009; Randrianasolo et al., 2014; Sun et al., 2017). However, forecasting of flood peak and timing is a different application to the forecasting of streamflow volumes. A number of data driven modelling studies have demonstrated that monthly streamflow lagged by one (or more) months provided some useful information for forecasting at a one month lead time (e.g. Bennett et al., 2014; Humphrey et al., 2016; Yang et al., 2017). This study demonstrates that these benefits also hold when CRR models, rather than data-driven approaches, are used as the forecasting model.*

3) Concerning the data assimilation and model updating issue, the bibliography is poorer (see detailed comment for page 3). Many references could be added. Since the authors chose to use the GR4J model and since the state updating they chose is the same one as the approach adopted for the GRP model ('adaptation' of the GR4J model for forecasting, used by the French flood forecasting centres), it is also worth mentioning this work (see detailed comments below). Beyond the references issue, it may (should ?) be noted that this study explores the benefits of model updating for seasonal forecasting, whereas many, if not most, studies consider shorter lead-times. This aspect has to be mentioned, since it is well known that the effects of model updating most often vanish whe the lead-time increases. In my opinion, keeping benefits at large lead-time is one of the (surprising) key result of this study and may be usefully emphasized.

See the above comment (2) for the changes due to the benefits at longer lead times. Thank you for the very constructive suggestions for relevant literature here and in the detailed comments. The literature review has been updated to include these references and points to improve the representation of the previous literature in the field of data assimilation.

*In CRR models, catchment conditions are represented by (usually multiple) model storages, referred to as "state variables". The values of these storages at the start of a forecast period are typically determined using a warm-up period, which allows the internal model states to reach reasonable values. Given the expected influence of the initial conditions on the simulated streamflow, observed data can be assimilated into the model to update the state of the model storages. The most commonly used approaches in hydrological data assimilation include direct updating of storages (for example Demirel et al., 2013), Kalman filtering, particle filtering, and variational data assimilation (see Liu and Gupta, 2007). Berthet (2010) considered a number of tests for different updating approaches for the GRP model, a CRR model commonly used in short term streamflow forecasting applications.*

4) A few methodological choices may deserve a little more discussion or explanation: The calibration algorithm is a rather complex one, but used with assumptions which are known to be not met in most cases (page 9, line 7: independent, homoscedastic residuals). Moreover, these assumptions are not consistent with the choice of the model error post-processor (a Box-Cox transformation is used in order to take into account the heteroscedasticity of these same residuals). Why did the authors pick a complex approach with unverified and inconsistent assumptions rather than a simpler one? It let the reader think that the authors used "components" available on the shelf or a pre-existing tool, which is quite understandable. But then they have to justify these choices (and why a so complex calibration method when much simpler ones are easily available?).

It was considered that a calibration method capable of estimating a posterior distribution of parameter values was required. This allowed the change in the distribution of suitable parameter values to be considered over time (Figure 8). In contrast, a simpler algorithm (gradient or evolutionary method) that provides a point estimate of the calibrated parameter values would not show the parameter identifiability (spread in values), and in turn the trends over time may not have been able to be separated from the variability in estimates. As such, the choice of algorithm is not considered overly complex, and a significantly simpler approach to estimating parameter uncertainty is not known to the authors.

The assumptions of the GR4J model likelihood function and the error model likelihood function are different, as they are applied in different situations. The following has been added to the manuscript.

*The assumptions of the post-processor residual error model used to estimate predictive uncertainty for monthly volumes are different to the assumptions of the residual error model used in the likelihood function for calibrating the daily GR4J model. As outlined in Section 3.2, the GR4J model is calibrated at the daily scale to observed streamflow using the standard least squares likelihood function, because it better captures the high daily flows, important for estimating the monthly volumes. The post-processing error model for the monthly volumes is designed to capture the predictive uncertainty in these monthly volumes, in particular the heteroscedasticity and skew of the residuals (McInerney et al., 2017; Refsgaard, 1997). These choices of residual error models at the daily and monthly time scales contribute to the study objectives of reliable forecasts at the monthly time scale, and are common in forecasting applications (for example, Lerat et al., 2015).*

5) Furthermore, one point is not discussed but may deserves some attention. Like the hydrological model, the model error post-processor is calibrated (not in a joint manner however). Why does the study on the impact of the calibration data period length on the calibration only focus on hydrological parameters and not on the post-processor parameters as well (mu, sigma)? This can indeed be treated independently (therefore not necessary in this article), but this research issue may be usefully mentioned. Are the post-processor parameters concerned by the rolling calibration?

The error model parameters are also determined using the rolling calibration approach, and this has been clarified in Section 3.5. This way any trends in the error model parameters are captured in the same way as those for the hydrological model. For the sake of brevity, a figure similar to Figure 8 has not been included for the error model parameters.

6) One element may also be better detailed: the results are given at a monthly scale (time step), whereas the GR4J model is a daily one: the way the GR4J model is run has to be precised. This is important, since the model is updated and effects of model updating decrease when the lead-time increases. However, it often does not only depend on the lead-time 'absolute' value but also on the number of time steps to reach this lead-time.

Good point. Clarification of the daily scale GR4J model have been added in Sections 3.1 and 3.3. The following has been added at the start of Section 3.5 to be clear when the monthly aggregation occurs (prior to the application of the post-processor error model):

*The monthly streamflow forecasts are obtained by aggregating the daily GR4J simulations. In order to quantify predictive uncertainty using a residual error model, the monthly-aggregated GR4J simulations, $Q^{\theta}$ , are compared to observed monthly streamflow volumes, $\widetilde{Q}$. The quantification of error is based on residuals errors, defined by the differences between observed and simulated monthly streamflow. Separate error models are estimated for the GR4J predictions for each catchment and for each type of forcing data (observed or forecast rainfall), as follows:*

7) In a nutshell, this article brings some interesting results, even in a field explored by many previous studies, and deserves publication. The suggestions made in order to improve the manuscript lead me to propose a moderate to major revision (however another round of submission afterwards does not seem necessary).

Thank you for the constructive comments that will help to improve the manuscript substantially.

**DETAILED COMMENTS**

- Page 2

8) Line 22 ("As these models are conceptual, they require calibration [...]"): they are not the only models that do so. Even the (so-called) physically-based models which could theoretically not need calibration, are most often calibrated, for various practical reasons.

Agreed, the manuscript has been updated as follows:

*The parameters of these models have a limited relationship to measureable catchment attributes (e.g. soil horizon depth) (e.g. Fenicia et al., 2014), and typically require calibration to observed streamflow data (noting that physical models also require some calibration (Mount et al., 2016; Pappenberger and Beven, 2006)).*

9) Lines 22 - 23: Brigode et al. (2013) show that this general a priori (longer calibration periods produce more robust parameters estimates) is not always verified. Therefore if this article is quoted (and it should be, in my opinion), it would be fair to indicate their results.

Agreed, the manuscript has been updated as follows:

*It is generally considered that longer calibration periods produce more robust parameter estimates, as a longer period exposes the model to a more diverse range of catchment conditions and flow events (Wu et al., 2013), however this is not always the case (for example Brigode et al., 2013).*

10) Lines 30 - 31: the definition of catchment non-stationarity which is proposed, is very interesting since it 'focuses' (restrains to) the physical object non-stationarity. However, it seems to be a binary state: the catchment is or is not stationary. Have the authors considered the notion of a "degree" of non-stationarity? Indeed, all the listed factors of non-stationarity are not expected to have the same consequences over the catchment behaviour. Might the rolling calibration approach be a tool to assess the relationship between "degrees" of non-stationarity and parameters evolution? (see also detailed comment on Fig. 3)

This is a good point, indeed natural systems are likely to be in varying degrees of non-stationarity, as nothing in nature is entirely static. The manuscript has been updated as follows:

*In this work, the term "non-stationary" is used to refer to situations where physical changes are expected to have occurred in a catchment, and where there is evidence to reject the hypothesis of stationarity. In practice, catchments may have different "degrees" of non-stationarity, depending on the evidence available to reject the hypothesis of stationarity, the degree of change in a catchment, and the time scales over which the changes take place.*

- Page 3

11) Line 1: is groundwater depletion a physical change (of the catchment) or the consequences of some of the listed catchment changes?

This has been clarified to refer to groundwater abstractions, as opposed to other potential causes of groundwater depletion:

*Examples of non-stationary changes in catchment conditions that could be expected to change the rainfall-runoff relationship include changes in land use or land-cover (e.g., deforestation, urbanization), land drainage, interception (e.g. dams or diversions), groundwater abstractions or responses to changes in climate (Milly et al., 2015).*

12) Lines 15 - 20: the bibliography review is rather poor: it gives some "extreme approaches" between the (too) simple GLUE and the very detailed BATEA (or similar approaches). Furthermore, GLUE is a quite old approach, giving a reference of 2008 is a bit strange

(unfair?), as it appears more recent than much more advanced and sophisticated approaches, as those developed by Kavetski, Vrugt and others. Since the chosen approach is a model error post-processor, I suggest Krzysztofowicz and Maranzano (2004).

Thank you for the reference, has been included and the paragraph edited as follows:

*Predictive uncertainty quantification is another major aspect of practical streamflow prediction. Many approaches are available to quantify predictive uncertainty, from approaches that identify a range of model parameters that represent the behaviour of the catchment using approaches such as generalised likelihood uncertainty estimation (GLUE) (Beven and Binley, 1992), to post-processor approaches (e.g. Krzysztofowicz and Maranzano, 2004), to disaggregation approaches that attempt to characterise each individual source of error explicitly (e.g. Kavetski et al., 2003; Vrugt et al., 2005). In this work, predictive uncertainty is estimated using an aggregated post-processor residual error model. The residual error model represents the differences between the hydrological model predictions and observed data, without trying to identify the contributing sources (Evin et al., 2014). The post-processor approach is chosen because it can lead to more robust estimates of predictive uncertainty compared to joint calibration of all parameters (i.e. estimating CRR model and error model parameters concurrently) (Evin et al., 2014).*

13) Line 17: "using a model error post-processor" rather than "using a post-processor error model" ?

The terminology has been clarified, see response at 12) above. "post-processor error model" is preferred by the authors, as the error model is developed and applied after the hydrological model has been calibrated, rather than calibrated in conjunction with the hydrological model (i.e. post-processor), and "error model" emphasises that an error model has been used.

- Page 4

14) Line 3 ("up to one month"): I understood this paragraph as a bibliography review giving general results (not specific to some catchments). However, it gives some values of the "influence duration" of the initial state, which strongly depends on the catchment characteristics. I am pretty confident in the fact that it easy to find catchments where the impact of the initial conditions is important during several months (even years).

This paragraph is indeed intended to review general results, and the "up to one month" statement is from the cited references (Li et al., 2009; Wang et al., 2011). However, it is agreed that this is not a universal rule, and will change depending on the catchment. The sentence has been changed to:

*Much of the skill in seasonal streamflow forecasts over periods following rainy seasons is commonly attributed to accurately representing initial catchment conditions (Koster et al., 2010; Pagano et al., 2004; Wang et al., 2009). In contrast, forecast skill over periods following dry seasons is generally attributed to both initial catchment conditions and meteorological inputs (Maurer and Lettenmaier, 2003; Wood and Lettenmaier, 2008). The impact of the initial catchment condition is particularly pronounced when forecasting over short lead times, typically up to one month (Li et al., 2009; Wang et al., 2011), although this time frame is generally catchment dependent.*

15) Line 5: "warm-up" rather than "warmup"?

The change has been included throughout.

16) Lines 14-16: it may be specified that this impact has been deeply evaluated for shorter lead-times (this emphasizes the character of novelty of the study). Furthermore, I disagree with the second sentence as it has been shown that the impact decreases quite fast (for many not too slow catchments) and is almost negligible at a seasonal scale (see e.g. Berthet et al. 2009 that the authors quote elsewhere). That is one very interesting aspect of the results of the submitted study.

This was the intent of this paragraph, to highlight that the impact at seasonal scales has not had extensive evaluation. The paragraph has been changed to:

*Studies on observed data assimilation and CRR model state updating have focused primarily on flood forecasting with short lead-times. The benefits at longer lead-times (e.g. seasonal) to forecast water availability have received less attention in the published literature.*

17) Line 18: I suggest to precise "calibration periods choice" or "calibration periods length" rather than only "calibration periods".

This is a useful clarification, and "Calibration Period Length (CPL)" is now used throughout.

18) Line 21: to enhance "seasonal" forecasting skill?

The objectives have been updated as:

*Evaluate the ability of state updating in a daily CRR model to improve predictive performance when forecasting streamflow volume for the upcoming month.*

19) Line 22: Does the article "demonstrate" that calibration period choice can affect forecast skill (that is quite known) or does it assess how much it does so?

The objective has been updated to "assess" rather than "demonstrate"

*Assess the degree to which using a shorter calibration period, that is more representative of the forecast period, can improve predictive performance, in particular when there is evidence of catchment non-stationarity.*

- Page 5

20) Line 11: is it the gauge "A2390514" rather than "A21390514"?

Correct, the typo has been corrected.

21) Lines 26 - 27: an hydrograph may be useful to support this information.

A boxplot of the monthly flow in Drain M has been included to demonstrate the variability (Figure 2).

- Page 6

22) Lines 3-13: is the description of the model developed by eWater Source useful for the reader. If I understood correctly, it is not directly related to the model used in this study. If so, this might confuse a bit the reader. E.g., I am not sure that the assumption of a constant inflow of salinity (which is not discussed) is needed by the reader to understand how the authors worked to answer to the scientific questions (which are the core of the article). The multi-objective nature of the calibration is also of no use for the rest of the study. If the fact that this model is used in practice had consequences on the methodological choices for this study, then the authors may consider explaining it (and discuss results with respect to it and to the specific context of wetland management).

As pointed out at 1), we agree that Section 2 includes superfluous detail that will be removed. This section on the separate water balance model in eWater Source has been removed altogether.

23) Line 14 ("To use this model for to inform operations"): it is always tricky for a non native English speaker to ask so to native ones, but may the authors check English here?

Correct, the typo will be corrected.

24) Line 15: "lead-time" rather than "leadtime"?

Correct, the typo will be corrected throughout.

25) Line 15: to fully understand the implication of the choice of the 1-month lead-time, it is necessary to know that the CRR model is a daily one (not only because the model is updated). However this information is given at subsection 3.1 (and not very explicitly: the reader has to know that GR4J is a daily model)

Good point, this information will be included as part of the aims and objectives in section 1 (see response 18).

26) Line 19 ("reasonable forecast skill is expected to be possible compared to longer forecast horizons"): may the authors provide some references? How much are the performances expected to decrease for longer lead-times? Furthermore, why did the authors choose to focus on a single lead-time? The evolution of the benefits of the model updating, with respect to the lead-time, in a context of seasonal forecast, would be a very interesting result.

We agree that this comparison of longer lead times, and how they influence the state updating performance, would be a very interesting study. However, it was considered beyond the scope of this work. The statement is based on the fairly obvious observation that the skill of rainfall forecasts is expected to reduce the longer the forecast horizon, and as such the skill of the streamflow forecasts will also reduce. This sentence has been removed.

27) Line 20 ("The mean annual rainfall for the region is in the range 600-675 mm"): page 5, lines 3 and 4 suggest some spatial variability. How strong is it? (600 to 675 mm is not very strong difference, compared to some other climates around the globe).

As pointed out, ~10% range is not very strong, but it is spatially consistent, as opposed to representing annual variability. The manuscript has been updated to clarify that there is a rainfall gradient from south to north:

*The mean annual rainfall for the region is in the range of 600 mm in the north to 675 mm in the south*

28) Line 20: is it useful to precise what "FAO56" stands for?

The acronym has been expanded and a reference included (Allen et al., 1998). It stands for Food and Agriculture Organization of the United Nations, and paper 56 relates to guidelines for computing crop water evapotranspiration.

- Page 7

29) Lines 3 - 4 ("2 rainfall hindcasts [...] were downscaled to the single rainfall gauge scale"): just to be sure, does it mean to the pixel where the gauge is?

That is correct. The pixel size is ~250 km, which tends to smooth out the rainfall events. The downscaling process, mapping the pixel where the rainfall gauge is to the gauge data, is used to restore more representative rainfall events. The manuscript has been modified as follows:

*POAMA-2 predictions have a coarse spatial resolution (~250 km), which does not capture the spatial variability in catchment-scale rainfall. For the purposes of this application, the POAMA-2 rainfall hindcasts (i.e. forecasts developed by applying the modelling system to the historical period) at the relevant pixel were downscaled to each climate station in the study region (Error! Reference source not found.) using the statistical downscaling method detailed in Shao and Li (2013). Further details of the downscaling approach are provided in*

*Humphrey et al. (2016).30) Lines 15 - 25: the authors may consider whether this paragraph would not be better written earlier (e.g. among the first paragraphs of section 2).*

Agreed, the paragraph on the catchments and where they flow will be moved to Section 2.1:

*Mosquito Creek flows into Bool Lagoon (Catchment C1 in **Error! Reference source not found.**, area 1002 km².). Drain M commences at the outlet of Bool Lagoon, and a large catchment flows into Drain M between Bool Lagoon and a diversion point at Callendale (Catchment C3 with an area of 2200 km²). Finally, the Drain M local catchment contributes flow downstream of the Callendale diversion point, flowing into Lake George (Catchment C2, area 383 km²).*

31) Line 20 ("It should also be noted that releases from Bool Lagoon [...]"): why is it important to understand this scientific study? (I worry about missing something useful for the interpretation of the results)

This will be clarified. The point being made was it is rare that the upstream catchment contributes to the downstream catchment, as releases from Bool Lagoon are rare. The conflicting water requirements for the case study site have been clarified in Section 2.2 as follows:

*Drain M serves multiple competing demands on the water resources available in this catchment system. These demands influence the decision to use the regulators along the system:*

  a) *Bool Lagoon has water requirements that influence releases from the lagoon into Drain M.*

  b) *Lake George has water requirements to maintain the estuarine ecology of the lake, to support its significance as a biological resource, and as a resource for recreational fishing.*

  c) *The ocean outlet requires some flow to prevent sediment from entering Lake George and to maintain connectivity to the sea (which allows fish movement and aids fish recruitment). However, high flows may impact on sea grasses, due to their low salinity and high nutrient load.*

  d) *The wetlands of the Upper South East to the north typically benefit from as much water as possible from the Drain M system.*

32) Lines 30 and following: are the details about the streamflow measurements devices useful?

These details help establish that any perceived non-stationary trends are unlikely to be due to streamflow instrumental measurements. The section has been shortened as follows:

*The identification of high quality data is important because biases and systematic changes in the measurement of hydrological data can significantly affect model calibration and lead to non-stationarity in the estimated model parameters (Westra et al., 2014). Analysis of the data and monitoring stations suggested that streamflow data uncertainty is expected to be low, given the regular cross sections of the weirs used for monitoring stage and upstream drains, and the high number of gaugings (between 78 and 166 flow gaugings at each flow station) available to develop stage-discharge relationships.*

- Page 8

33) Line 3: I agree with the fact that indicating the data are of good quality and too often not done, but if the authors want to demonstrate the quality of the rating curves, they may add some information about the number of years during which the 78 and 166 gaugings have been achieved and how much often the rating curves have been modified.

See response to 32)

33) Lines 9 - 20: since catchment non-stationarity is an important issue for this study, I suggest to make this paragraph a subsection dedicated to this topic (here).

This information has been highlighted earlier in section 2.1 as follows:

*In the region where the case study catchments are located, plantation forestry expanded substantially in the late 1990s. Changes in the relationship between rainfall and runoff also occurred during this period, evidenced by the reduced slope in the plot of cumulative runoff against cumulative rainfall (double-mass analysis) in **Error! Reference source not found.** (Searcy et al., 1960; Yihdego and Webb, 2013). The runoff ratio in catchment C1 is approximately 0.045 before year 2000, but reduces by 70% to 0.013 after 2000. The runoff ratio in catchment C2 is around 0.088 before year 2000, but reduces by 30% to 0.061 after 2000. This comparison provides stronger evidence of non-stationarity in catchment C1 than in catchment C2. Other studies have also investigated the link between changes in the hydrology and changes in land use in the region (Avey and Harvey, 2014; Brookes et al., 2017). These changes have implications on the choice of calibration data period, as data from the 1970s may not be representative of hydrological conditions in the 2000s.*

34) Lines 23 - 24 ("GR4J [...] explicitly accounts for non-conservative (or 'leaky') catchments"): I agree. However, it should be kept in mind that GR4J has not been designed nor is known to achieve good performances for karstified catchments (mentioned page 7, line 13). Moreover, I am not convinced it is quite appropriate for ephemeral catchments (as suggested by line 21, page 11).

It is agreed that GR4J may not be ideal in ephemeral catchments, as the exponential decay relationship used in the storage reservoirs cannot completely dry out. However, this is a relatively theoretical consideration, for example, if any simulated flow below that which could be adequately measured is considered to be zero, ephemeral behaviour can be represented. As outlined, previous studies have

demonstrated good performance for Australian conditions, including ephemeral catchments (Coron et al., 2012; Guo et al., 2017). Westra et al. (2014) has also been added to this list, who applied GR4J in a similar location in southern Australia. Considering alternate model structures was beyond the scope of the study, however, it is possible (likely?) that more appropriate model structures could be identified in future work.

35) Lines 28 - 31: may the authors explain what motivates their choice of adding a 5th free parameter to calibration? Is it important for their particular catchments or for their methodology in this study?

As noted in the previous comment, GR4J was not designed for this application. The catchments considered have a relatively slow response, and it was considered that the pre-specified split to the routing store of 0.9 may be too low for these catchments. As can be seen in Figure 8, this turned out to be the case for the calibrated parameter values, where higher values of the split parameter were found, in particular for C2. This point will has been clarified in the manuscript as follows:

*Note that the catchments considered have a relatively slow streamflow response. Consequently, the pre-specified split to the routing store of 0.9 in the original specification of the GR4J model may be too low for these catchments. To mitigate this potential deficiency, we have modified the GR4J model so that the split between the routing store and the direct runoff is included as an explicit calibration parameter termed split.*

- Page 9

36) Lines 10 - 12 ("this function [RMSE] provides a focus on the highest flow in the time series, where the majority of the runoff occurs"): it is not necessary. It provides a focus on the largest absolute errors, which indeed most often occur for the largest flows. However, consider a hypothetical model whose errors would be only on low flows.

We agree, the original statement was not strictly correct and has been removed.

37) Line 26 ("External influences include model structural limitations [...]"): this confused me, after reading the catchment non-stationarity given on pages 2 & 3. Does it suggest that parameters variation due to structural deficiencies would be considered here?

We agree this sentence does not add value to the discussion in Section 3.3. It will be removed to avoid confusion.

- Page 10

38) Lines 3-5: Would not it be useful to emphasize the trade-off between a longer calibration period to reduce the parameter uncertainty and a shorter calibration period to mainly take into account the most recent dynamics in the introduction section?

This trade-off has been highlighted in two places in the revised manuscript:

Section 1:

*Often there is a trade-off between the benefits of a longer calibration period which exposes the model to a more diverse range of conditions and tends to improve parameter identifiability, versus the benefits of a shorter calibration period that exposes the model to the most recent – and hence often the most relevant – dynamics in the catchment. Demonstrating and understanding the impact of this trade-off on model predictive performance is a key research gap pursued in this study.*

Section 3.3

*Calibration period lengths of CPL = 10 years and CPL = 20 years length are considered, to assess the trade-off between using a longer calibration period to expose the model to more diverse catchment conditions and improve parameter identifiability, versus using a shorter calibration period length to expose the model to more recent hydrological dynamics.*

39) Line 13-14: the literature review is also poor about data assimilation and model updating. Generic references may be Refsgaard (1997) and Liu and Gupta (2007). Since the chosen updating approach is the same as the one used for the GRP model (which is a mere adaptation of GR4J for forecasting purposes), I suggest to refer to the work of the team which developed these models. The authors may pick Tangara (2005) and Berthet (2010), both in French, which described the numerous tests of different updating approaches made by the GR4J research team (some of them discussed in section 5! See comment below) and detailed the resulting GRP model. They may prefer Berthet et al. (2010), which provides a much shorter description of the model and the updating techniques but also discusses the impact of the largest errors on the RMSE-based criteria values (see discussion page 9). For a detailed description of the GRP model, the authors may also consult: https://webgr.irstea.fr/en/modeles/modelede-prevision-grp/fonctionnement-grp/. Moreover, since sequential approaches such as ensemble Kalman filter and particle filters are mentioned, I suggest also to add references to Moradkhani et al. (2005, 5005b) and Weerts and El Serafy (2006).

Thank you for the very constructive suggestions. The literature review has been updated to include these references, in particular Lui and Gupta (2007) as a review of data assimilation approaches:

*In CRR models, catchment conditions are represented by (usually multiple) model storages, referred to as "state variables". The values of these storages at the start of a forecast period are typically determined using a warm-up period, which allows the internal model states to reach reasonable values. Given the expected influence of the initial conditions on the simulated streamflow, observed data can be assimilated into the model to update the state of the model storages. The most commonly used approaches in hydrological data assimilation include direct updating of storages (for example Demirel et al., 2013), Kalman filtering, particle filtering, and variational data assimilation (see Liu and Gupta, 2007). Berthet (2010) considered a number of tests for different updating approaches for the GRP model, a CRR model commonly used in short term streamflow forecasting applications.*

40) Lines 17 - 27: a flowchart would greatly help the reader.

It is considered that the equations provided are the clearest approach to explain the exactly methodology used to update the routing store to simulate the observed flow. Further details on this approach are provided in the original work, Demirel et al. (2013). This link to the earlier work has been made clearer in the revised manuscript.

41) Line 27: "where X3 is the estimated runoff model parameter" rather than "where X3 is an estimated runoff model parameter"?

This change has been made.

- Page 11

42) Lines 3 - 4 ("particularly when used to update both model state variables and model parameters"): I agree with the authors, but is it relevant here? (since parameters are not updated here).

It is agreed that this is out of place and has been removed.

43) Line 12 ("Depending on the case"): this is not clear, until the reader reaches section 3.7.

"Case" and "model configuration" has been clarified and made consistent throughout the manuscript, and any forward referencing removed. The clarification is as follows:

*Two options for state updating (with versus without) and two options for calibration period length (CPL = 10 years versus CPL = 20 years) are considered. The combination of these options leads to four model configurations. Four different cases are considered for each model configuration, given by the combinations of two catchments (C1 and C2) and two sources of climate data (observed and forecast). This results in a total of 16 scenarios considered.*

- Page 12

44) Line 5: why do the authors prefer to sample the (normalized) residuals rather than picking a number of calculated quantiles (from the Gaussian distribution)?

A direct selection of quantiles would provide an analytical approach to represent the distribution. However, this can be difficult, as the calculated quantiles in normalised space do not correspond to the same quantiles in un-normalised space, due to the combination of the transformation, the truncation of very low flows and the parameter uncertainty. The Monte Carlo Simulation approach used is more computationally intensive, but the most robust. This approach is also more generalised, and would be required is more complex error models were used. For these reasons the approach used has been retained.

45) Line 18: may the authors explain the choice of the 0.05 and 0.95 as normalized extrema values?

This range was adopted for data visualisation purposes only so that the worst-performing case for each metric had some area shown in the plot. However, it is agreed that this was confusing, and a 0-1 normalisation is now used in Figure 4.

- Page 13

46) Lines 17 - 21: as pointed out by the authors, the reference distribution has an 'unfair' advantage. Then may the authors explain this choice? Why have they not chosen a simple naive forecast model?

The reference distribution of the monthly streamflow is considered a simple naïve forecast model. Other approaches could be used, autocorrelation with last month's streamflow, for example. The reference to an unfair advantage has been removed, as it is expected that the forecast models tested should be able to perform better than an uninformed climatology, and as such is considered a useful baseline for the calculation of forecast skill. The paragraph has been changed to:

*The reference distribution for each month is calculated as the empirical distribution of all observed data in that month, using the entire set of observed data (including data from the prediction period). This approach provides a stringent baseline for the CRPS normalization in Eq. (13).*

47) Line 23: check the formula. There is missing sum for the denominator.

This typo has been corrected.

- Page 14

48) Line 6: is the rainfall forecast used here the ensemble forecasts described in subsection 2.1? I don't think it obvious. If not, why were the ensemble described?

Yes, the ensemble forecasts were used in all cases. This has been clarified in Section 3.5:

*When forecast rainfall is used as input to GR4J, an ensemble of daily streamflow forecasts is produced (with a single GR4J streamflow time series per rainfall forecast time series). Each such "individual" daily GR4J time series is then aggregated to a monthly time step. The time series $Q^\theta$ is constructed from the time series of medians of the individual monthly streamflow time series. This use of the median streamflow forecasts from the multiple ensembles of the meteorological ensemble forecasts may result in some information loss, but aggregation approach of streamflow forecast ensembles is commonly used in operational applications (e.g. Lerat et al., 2015; Matte et al., 2017; Schepen et al., 2017; Wani et al., 2017). Further work is needed to more fully utilise the information from ensemble forecasts when developing post processing models.*

49) Line 6: I found only 2^4 = 16 cases (model with or without updating; 2 calibration period lengths; 2 catchments and 2 rainfall forcings). What do I miss?

Well-spotted - this is an error and will be corrected to 16 (see response 42).

 - Page 15

50) Line 6 ("Any detrimental impacts"): check English.

The sentence has been removed.

51) Line 8 (and followings): I suggest to precise "calibration period length" rather than only "calibration period"

This is a useful clarification, and "Calibration Period Length (CPL)" is now used throughout.

52) Line 16: the differences are not much smaller for catchment C1. How much are they significant?

This section has been largely rewritten in the revised manuscript. Significance tests have not been calculated due to the lack of replicates.

53) Line 19 ("the differences were more pronounced for the most practically relevant cases with forecast rainfall [...]"): this is of particular interest for operational purposes (e.g., forecasts) and is worth being emphasized.

On review of this conclusion, it was found that this was in part due to the different time periods used for the observed and forecast rainfall cases (as outlined in Section 3.7 previously). The observed rainfall results have been recalculated over only the period used for the forecast rainfall case, and now the results are consistent between the two rainfall cases.

54) Line 20: I don't understand how the model error post-processor compensates the introduced errors. I thought that it only assesses them.

The normalised residual can have a non-zero mean, which can compensate for biases in the hydrological model predictions. But the main purpose of the error model is to quantify the uncertainty in forecasts, not to compensate for errors. For clarity, this point has been removed.

55) Lines 23-24 ("catchment C1 had been identified to have a substantial reduction in the rainfall-runoff relationship over time"). As discussed below (comments on Fig. 3), the catchment appears as rather stationary up to 1990 (approximately) and then also more or less stationary from 1990 to 2010. If it is so, how can the difference in calibrated parameters obtained with the 2 different calibration period lengths for years 2009 2010 be explained by this change around 1990?

The 20-year calibration period was 1989-2008. The 10-year calibration period was 1999-2008. As such, the early 1990s are part of the 20 year calibration period, but not of the 10 year calibration period. The results indicate that the 20 year calibration period resulted in model parameters that produce more flow the 2009 (and 2010) validation year than the 10 year calibration period. This suggests that streamflow data from the 1990s are substantially different from streamflow data from the 2000s. The manuscript will be changed as follows:

The differences between the streamflow predictions obtained in the two catchments C1 and C2 (for the case of GR4J forced with observed rainfall) are illustrated in Figure 7 for the most recent period 2009-2011. In catchment C1, using a longer calibration period length tends to yield wider prediction limits and an overestimation of the observed flow in 2009 and 2010, whereas using the shorter calibration length provides a better capture of the catchment response in these two years. In contrast, in catchment C2, which has less evidence of non-stationarity (Section **Error! Reference source not found.**), the calibration period length makes very little difference on the resulting streamflow predictions.

- Page 16

56) Line 16 ("A model fitting anomaly resulting from a shorter calibration period"): did the author investigate this "anomaly"? How can it be explained?

This was not investigated further. It is assumed to be due to reduced parameter identifiability. A different combination of parameters (e.g. higher X4 and lower X2 and split compared to the periods before and after) resulted in similar values for the objective function for this particular period. The manuscript has been changed as follows:

An exception to the pattern of the median parameter values being insensitive to calibration period lengths can be seen in 1999, where the use of the 10 year calibration period length produces higher values of X4 and lower values of X2 and the split introduced in this study (Section 3.1). This exception could represent a model fitting anomaly resulting from a shorter calibration period length.

57) Line 26: this is interesting at a seasonal scale, since it has been shown that hydrological models are "stable", i.e. the updating effect vanishes after a number of time steps (e.g. Berthet et al. 2009 at a hourly time step: then after a few days at most for a large majority of the tested watersheds).

Thank you. This point has been added to section 5.1 (see response 2)

58) Line 29 ("As the range in model predictions should be reduced by forcing the model to simulate the observed streamflow at the start of the forecast period"). Is there any confusion between precision and sharpness? Model updating increase sharpness and precision (at least for the shortest lead-times).

The term "precision" was used in this work as a synonym for "sharpness", as outlined on Page 12 line 24. However, as "sharpness" is a term more generally used in the foresting community, "precision" has been changed to "sharpness" throughout.

59) Line 30 ("the trade-off for an increase in precision would typically be a reduction in the reliability of the predictive uncertainty"): again I assumed that the authors meant "sharpness". I suggest to write that this trade-off for an increase in sharpness ** may ** result in the reduction of the reliability, if the authors do not provide a general (theoretical) explanation. As much as I know, this is a common feature, but exceptions should exist, and the last sentence of the paragraph (page 17, lines 1 - 2) says so.

See above, "precision" will be changed to "sharpness" throughout. Nonetheless, this is a good point, as the result cannot be proven to be generic. The qualification "may" has been added to this sentence.

- Page 17

60) Line 4 ("update the GR4J production store along with the routing store"): this has been tested by the GR4J team (Berthet, 2010), with no significant improvement in a forecasting context at a hourly time step.

This point has been removed, as on review it was considered not directly relevant to the main results presented.

61) Lines 4 - 5 ("This could be expected"): may the authors give some explanation to found this idea?

See response to 60)

62) Lines 3 - 9: this paragraph is very interesting, but how is it related to the scientific issues developed in this article?

See response to 60)

63) Lines 16 - 19: in my opinion, this is the (or one of the) key findings. How may the authors emphasize it, rather than putting it at the very end of the article?

See response to 53)

64) Line 30 ("in most cases"): the study was driven only on 2 catchments... It should be pointed out that a work over a (much) larger number of catchments is needed to ensure the generality of these interesting results.

See response to 60)

- Page 19

65) Tab. 1: the authors may usefully add the parameters meanings and their units (and a GR4J flowchart aside).

Parameter meanings and units have been included. A GR4J flow chart was not included for brevity, with the reference to Perrin et al. (2003) use for this purpose.

- Page 20

66) Fig. 1: the drain M is not given in the legend and has the same color as catchment boundaries, which makes it difficult to identify.

Good point - the map has been adjusted accordingly.

67) Fig. 2: what are the upper and lower bounds? Minimum and maximum of the ensemble? Some predictive quantiles such as 0.05 and 0.95? If the latter, is there any information about the reliability?

This figure has now been removed as it did not directly contribute to the results presented.

- Page 21 (Fig. 3)

68) I wonder if there is not a "sudden" change around 1990: catchments C1 and C2 look quite stationary up to 1990, and also after 1990. If it is so, can it really be explained by plantation forestry expansion (which is more a "continuous" factor of non-stationarity). Furthermore, can this question the relevancy of the rolling calibration which might be better adapted for smooth non-stationarity?

A step change is arguably more likely to have taken place over the mid-late 1990s. The early 1990s are also quite wet, which may result in steeper slopes for these years. A cause for a sudden change in the streamflow response around 1990 is not known to the authors. Reductions in groundwater levels due to the forestry expansion has been attributed in other studies (Avey and Harvey, 2014; Brookes et al., 2017).

In any case, identifying such a distinct step change, or even a continuous change, is very difficult to do with certainty. The difficulty in identifying such step changes, and the idea of using the rolling calibration approach as a means to overcome this, this has been added to Section 3.3:

*This methodology allows the identification of changes in parameter distributions over time, without the need to identify specific periods when changes in the rainfall-runoff response may have occurred.*

- Page 22 (Fig. 4)

69) It is very important to insist on the fact that the plot gives relative values of the metrics (higher is better) which are then not consistent with the formulas and details given in subsection 3.6. I was first confused becaused I did not notice (at first) the mention in the y-label (even if I admit that it is written in (sufficiently) large characters). I suggest to change the criteria described in section 3.6 to give only their relative values.

The following has been added to Section 3.7:

*To ensure a consistent comparison of multiple model scenarios, the metrics are computed as follows:*

- *the same period is used to calculate the metrics in all cases. This period was determined by the availability of the forecast rainfall, from May 2001 to April 2011.*

- *the performance metrics are normalized by linearly scaling the worst value to a value of 0 and the best value to 1,*

$$M_r = \frac{M - M_w}{M_b - M_w} \qquad\qquad (1)$$

*where the worst and best values for each metric, $M_w$ and $M_b$, respectively, are listed in **Error! Reference source not found.**. The remainder of the presentation, in particular **Error! Reference source not found.**, reports the normalized metrics computed using Eq. (15).*

- Page 23 (Fig. 5)

70) Why are the results plotted only up to 2005?

This was done for illustrative purposes only. Some of the desirable detail in the hydrographs is difficult to see when the whole time series is plotted (~20 years). As such, a subset of the data was shown, to provide a trade-off between showing all of the results, and making a clear point. The full plot has been provided as supplementary material to show this information.

Page 25 (Fig. 7)

71) Why are the results plotted only from 2008 to 2010?

See response to 70).

*Response to Interactive comment on* "State Updating and
Calibration Period Selection to Improve Dynamic
Monthly Streamflow Forecasts for a Wetland Management Application" *by Matthew
S. Gibbs et al.*

**Anonymous Referee #2**

General:

The manuscript of Gibbs et al. evaluates the effect of calibration setup on the GR4J model performance within 2 Australian basins on 1-month lead-time hydrologic forecast. Authors draw mainly following conclusions based on 2-basin analysis using 5 indicators: (I) the length of calibration period does not necessarily should be as long as possible, in particular when changes in flow regimes are observed. Additionally, (II) the authors state that a simple model state updating improves hydrological forecast at 1-month lead time.

Based on my review, I consider the overall topic to be relevant for HESS, however, some parts of the manuscript need to improved and clarified, as further suggested below.

*Thank you for the constructive comments that have improved the clarity and contribution of the manuscript.*

Major comments:

1. In general, it is not surprising that updating model initial conditions has benefits on hydrologic forecast. Additionally, it is not surprising that changes of physiographic conditions may change the catchment's response, indicating different information content/validity of observed discharge data on model parameters. This only confirms observations of previous studies, which some of them are cited. Would be nice to more clearly demonstrate benefits over existing/operational approaches (in terms of costs etc). In particular, when the title includes words like "Management Application".

*Commentary on the benefits over the existing approach has been added in Section 5.3:*

*The forecasting approaches developed in this work can support improved water management in the drainage system considered. The approach currently used by the management authority is very conservative: streamflow forecasts are not attempted, and changes in water management are made only once downstream requirements have been met. With the forecasting models and methods developed in this work, it becomes possible to produce streamflow forecasts with a high reliability, improved sharpness and reduced bias. Thus it becomes possible to provide useful probabilistic estimates of how likely it is that the downstream flow requirements will be met in the next month. With this information, managers can more confidently consider increasing the frequency and duration of inundation for many of the wetlands in the region, and can make decisions on management changes much earlier in the season.*

*The Discussion section has also been expanded to include Section 5.1 to highlight the key findings over previous approaches:*

*Most previous studies have used state updating in a short term flood forecasting context, and found limited effect of the initial conditions after a number of days (e.g. Berthet et al., 2009; Randrianasolo et al., 2014; Sun et al., 2017). However, forecasting of flood peak and timing is a different application to the forecasting of streamflow volumes. A number of data driven modelling studies have demonstrated that monthly streamflow lagged by one (or more) months provided some useful information for forecasting at a one month lead time (e.g. Bennett et al., 2014; Humphrey et al., 2016; Yang et al., 2017). This study demonstrates that these benefits also hold when CRR models, rather than data-driven approaches, are used as the forecasting model.*

2. Unfortunately, the analysis is limited to two basins, which really can't be used to draw any conclusions (as authors also recognise in the end). I would strongly encourage authors to enlarge the number of basins and events. Another two basins may yield completely different results; therefore, generality should be avoided.

*It is agreed that general results cannot be drawn from the results based on two basins. The drainage system considered was based on a user need for seasonal forecasts, which resulted in the limited application to two basins. The Section 5.4 has been included to outline this important limitation:*

*The enhancements to predictive performance of streamflow forecasts from state updating and a shorter calibration period have been demonstrated on two catchments. These catchment were selected based on an established user need for seasonal forecasts to improve the water management of a channel drainage system with multiple competing demands. Importantly, the case study catchments in this work are ephemeral and dry, with low runoff ratios. These types of catchments are known to be challenging to model (McInerney et al., 2017; Ye et al., 1997). Future work will evaluate the proposed seasonal streamflow forecasting techniques over a wider range of catchments and environmental conditions.*

3. Would be nice to relate your results with another study, which details CRR state updating for 1-month forecast. However, I wonder, whether it is really the effect of state updating here. It is well recognized that the effect of initial conditions (based on discharge observations), in such small basins, diminishes after a couple of days. Please, comment on this.

The result that model updating still improved forecast performance at the lead-time of one month is considered one of the key results of this work. The controlled study for testing the model with and without state updating for all the cases considered (observed and forecast rainfall, two lengths for the calibration period and two basins), demonstrates that for the scenarios considered the effect can be attributed to state updating. The point at Response 2, on the potentially limited generality of this result, is relevant and has been included.

It is agreed that the majority of the literature focused on short term flood forecasting has found limited effect of initial conditions after a few days (e.g. Berthet et al., 2009). However, forecasting of flood peak and timing is a different application to that considered here, forecasting water volume available for environmental use. See response 1. for the inclusion of this point in the manuscript.

4. The description of basin and data is way too long and detailed (3 pages), in particular when the discussion and conclusion do not come to those details at all.

Reviewer 1 raised a similar issue, and as such Section 2 has been shortened to be more targeted and remove surplus detail One of the topics of interest for the sub-seasonal to seasonal hydrological forecasting special issue was user needs for seasonal forecasts. As such, more detail than typical on the case study application to wetland management was included in the manuscript. However, it is agreed that this is a distraction from the more generic scientific issues of interest to most readers. As outlined at Comment 1, commentary on the benefits over the existing approach has been added at Section 5.3.

5. Please, place error bars into figure 4, in the same way as the uncertainty is presented in following figures and provide discussion.

The uncertainty in the streamflow time series had been produced and assessed (e.g Figures 5 & 6) because it is a direct of the output of the modelling setup. However, all of the streamflow replicates are used in the calculation of the performance metrics (reliability, precision and CRPS in particular), and as such the uncertainty in the values of the metrics is not easily derived. To the authors' knowledge, it is no common to provide the uncertainty in performance metrics related to streamflow forecast uncertainty (e.g. Alfieri et al., 2014; McInerney et al., 2017; Pappenberger et al., 2015; Renard et al., 2010).

We do acknowledge that there is uncertainty in the values for the metrics, due to finite number of replicates and the finite length of the observed data (McInerney et al., 2017) and that in the presence of strong autocorrelation in the streamflow error time series this uncertainty may be significant. However, to develop an approach to quantify the metric uncertainties is beyond the scope of this study. Instead, we follow the guidance provided by McInerney et al. (2017) who suggested that these uncertainties are quite small and are unlikely to impact on the conclusions of the study. We do acknowledge that further work is needed in this area.

6. Discussion about alternative types of observations (besides Q) may be provided in the manuscript.

The literature review has been expanded as follows to refer to previous studies that have used other types of observations that could be used for state updating (e.g. either remotely sensed or ground based soil moisture), and why streamflow was focused on in this study.

*Updating the states of conceptual rainfall-runoff models is not straightforward, as any environmental model is at best an approximate representations of the real catchment (Berthet et al., 2009). A number of observed data sources can be used to update model storages, including observed streamflow and in-situ or remotely sensed soil moisture. From these options, Li et al. (2015b) suggests that gauged discharge data assimilation is a more effective way to improve short-term forecasts and is still preferred for operational streamflow forecasting purposes.*

7. What is the main applicability of your findings? Are they going to be used operational, if yes, what are the benefits over existing forecast method? Please, clarify.

See response to Comment 1.

Minor:

• Third sentence from the Introduction regarding Drain M catchment should be moved somewhere towards the end of Introduction.

This change has been made.

• P 6, L.5: evapoconcentration => evapotranspiration?

This typo has been corrected

• Section name 2.2 "Streamflow and streamflow data": sounds a bit repetitive

The section has been changed to "Streamflow data"

• P 9, L.21 "burn-in" into quotes

This change has been made.

• Eq. 12 is wrong, sum in the denominator is missing

Correct, the equation has been updated.

• Caption of figure 2: "POAMA" => "POAMA-2"

This change has been made.

*Interactive comment on* "State Updating and
Calibration Period Selection to Improve Dynamic
Monthly Streamflow Forecasts for a Wetland Management Application" *by Matthew
S. Gibbs et al.*

**Anonymous Referee #3**

This manuscript presents an original study focused on two important aspect of monthly streamflow forecasting: state updating and the selection of an appropriate calibration period. The watershed on which the methods are implemented and tested is a very interesting case study. This semi-arid watershed is extensively impacted by human intervention, as it includes diversions and "Drain M", so runoff water can be directed either toward the north to benefit wetlands or toward the south to maintain fish ecosystems and comply with other constraints. Adequate management of this watershed appears very difficult and it is evident from the authors' description of the situation that monthly hydrological forecasts are essential. The authors show that selecting a calibration period during which the conditions are close to the expected forecast conditions can improve forecasts performance substantially.

In my opinion, this is a well-written paper (very clear!) that brings interesting novel knowledge in the field of ensemble monthly streamflow forecasting. I also find it completely appropriate for publication in HESS, especially since the case study raises issues regarding water management and tradeoffs between ecosystem services and human activities.

I appreciate that the authors included a short discussion about the uncertainty related to the gauging measurements (page 8 first paragraph). I also like the idea of adding the "split" parameter to GR4J in the calibration process.

The conclusions drawn by the authors regarding the tradeoff between the length of the calibration database versus its "representativeness" is interesting. I appreciate that they recognize that said conclusions might be limited to their specific case study and that further investigation for a wider range of hydro-climatic regimes would be needed. I only have very few minor comments and suggestions that I think could improve the paper. I strongly recommend that it be published in HESS.

Thank you for the constructive comments that have improved the manuscript.

*Minor comments*:

1. Why did you consider only the median of the hydrological model predictions rather than the whole ensemble?

This question kept bugging me all along while I was reading. You have access to meteorological ensemble forecasts (page 6 line 30 to page 7 line 5). They are expected to account for the uncertainty related to the meteorological conditions (or at least a part of this uncertainty). Why then did you not keep all the scenarios after passing everything through the hydrological model? Is it to make it more comparable to the case where the hydrological model is forced by (deterministic) observations? If so, I personally don't see why this would be necessary. And then after, you use a statistical method to dress the median back to an ensemble. Is it because you found out that hydrological ensemble forecasts built only from meteorological ensembles were underdispersed and would have needed post-processing?

In my opinion, those choices (i.e. using the median, thus ignoring the other ensemble members, and then dressing the median into an ensemble) really need further explanations/justifications.

The focus of this work was to investigate the impact of the state updating and calibration periods on streamflow predictive uncertainty, and a current best practice post-processing approach was adopted to estimate this predictive uncertainty. It is common practice to use deterministic inputs, often based on summarising an ensemble of forecasts using summary metrics (e.g. mean or median) as the input to post-processing methods (e.g. Lerat et al., 2015; Matte et al., 2017; Schepen et al., 2017; Wani et al., 2017). As such, it was considered beyond the scope of this work to also consider improving methods for applying post-processing models.

However, it is acknowledged that that by only considering the median streamflow simulated across the forecast rainfall ensemble that some information is lost. The paper has been modified as follows:

*When forecast rainfall is used as input to GR4J, an ensemble of daily streamflow forecasts is produced (with a single GR4J streamflow time series per rainfall forecast time series). Each such "individual" daily GR4J time series is then aggregated to a monthly time step. The time series $Q^\theta$ is constructed from the time series of medians of the individual monthly streamflow time series. This use of the median streamflow forecasts from the multiple ensembles of the meteorological ensemble forecasts may result in some information loss, but aggregation approach of streamflow forecast ensembles is commonly used in operational applications (e.g. Lerat et al., 2015; Matte et al., 2017; Schepen et al., 2017; Wani et al., 2017). Further work is needed to more fully utilise the information from ensemble forecasts when developing post processing models.*

2. There are a couple of (very minor) elements that could be clearer

-        Page 5 line 30: please add a reference for the Ramsar list. I didn't know this list before reading the manuscript so I looked it up on the web. I think that a reference would be helpful to be sure that other readers like me know what you are talking about.

The reference Matthews (1993) has been added.

-       Page 9 line 21: What is a "burn-in" period? At first I thought it was a synonym of "warm-up period" in the context of the DREAM algorithm, but I am really not sure.

"burn-in" is the term commonly used in Markov Chain Monte Carlo modelling and is preferred to be maintained. Both "warm-up" and "burn-in" refer to the period at the start of the analysis that is discarded and not considered further, but the applications are different and as such the different terminology considered necessary.

3. There are a few typos in the manuscript and I am unsure of the spelling for 1-2 words:

- Page 3 line 27-28: parenthesis typo, replace "(McInerney et al, 2017)" by "McInerney et al. (2017)"

The typo has been corrected.

- Page 11 equation (5): Something seems wrong with the curly brace

The curly brace is commonly used notation for conditional expressions, in this case to avoid dividing by 0 when λ=0.

- Page 11 line 1: I think that the sentence "(…) are available, for example, the ensemble Kalman filter (…)" should be split, as in: "(…) are available. For example, the ensemble Kalman filter (…)"

The sentence has been largely reworded.

- Page 11 line 12: day "n" and month "t" should be in italics.

The change has been made.

- Page 12 line 17: I think that "straightforward" should be written in one word.

The change has been made.

- Page 13 equation 12: A summation seems to be missing at the denominator. Also, I don't think there should be a "t" index for the average streamflow, since by definition it is independent from time (it is the average of the time series).

The summation typo has been corrected. It is agreed that the index t for the average is also incorrect. Thank you for picking these up.

- Page 14 line 6 (and several other places in the manuscript): Why do you write "observ*ed* rainfall" but not "forecast*ed* rainfall" I am not a native English speaker so perhaps I am completely wrong, but I could not help but finding this strange.

This is a strange nuance of the English language. Both "forecast" and "forecasted" can be used as past tense for the verb forecast. It was considered that "forecast" is more commonly used in the literature and is proposed to be maintained throughout.

- Page 14 line 16: Is there a "to" missing in "The changes due adopting (…)"?
The typo has been corrected.

[revised manuscript text omitted]

---

## Author Response (AR2)

**Response to Editor Decision:**

**Publish subject to technical corrections (11 Dec 2017) by Maria-Helena Ramos**

I read carefully your answers to three reviewers and your revised version of the paper. I agree with the reviewers that the paper is worth publishing and I consider that you have addressed well the remarks and suggestions from the reviewers, which greatly improved your manuscript.

Thank you, we agree the reviewer's comments have greatly improved the manuscript.

I have just a few remarks that I think would be interesting to be considered for the final paper (I consider them just technical remarks with no need for a new round of revision, but I would appreciate if you could consider addressing them). They are listed below.

**Remarks:**

1) I agree with the first remark of reviewer #3 about using a single-value forecast to post-process streamflow predictions. Technically, ensemble dressing with model errors can consider all members of an ensemble without the need for reducing the ensemble to a mean or median value. I can understand your choice not to use the full ensemble members, but I have more difficult to find your arguments convincing. I think the sentences on lines 4-8, Page 10, need some rethinking. I do not think that the applications you cited for aggregating ensemble predictions are used in the correct context. The study of Matte et al. (2017), for instance, focused on a utility model, not a post-processing development approach. Besides, the ESP approach is common in operational seasonal forecasting for water management and it is based on ensembles of historic precipitation used as input to a hydrological model. I would consider writing this sentence differently: "Although the use of aggregation approaches for single-valued streamflow forecast from ensemble predictions has been seen in operational applications (e.g., Lerat et al, 2015…), we note that this approach may result in some information loss."

Then, the sentence on lines 7-8 also raises a question: do you mean further work is needed within your study or in general? I think you could have considered the full ensemble with further work within your study, but I also think that we can find ensemble post-processors in the literature, so it is not as if there is nothing done on the topic. Post-processors that use the full information from ensembles do exist. I suggest that you either delete or you explain better what you mean with this sentence.

These are suggestions to clarify your opinion on the topic as expressed in your paper, so the reader can better apprehend your viewpoints.

Thankyou for the suggestions to clarify this point. The intention was to reflect further work in general could be undertaken to provide clarity on the best approaches to implement ensemble post-processors. However, as pointed out, studies that do use the full information from ensembles do exist. As such, this sentence suggesting further work has been deleted.

Lines 4-8 page 10 have been revised to the constructive suggestion made. Only the Lerat et al, (2015) reference has been retained. Along with Matte et al. (2017) pointed out above, the two other citations used a single streamflow forecast to calibrate and apply a residual error model for streamflow forecasts, but there was not a preceding step of aggregation of an ensemble of streamflow forecasts. The sentence has been changed as follows:

*Although the use of aggregation approaches for single-valued streamflow forecast from ensemble predictions has been seen in operational applications (see, for example, Lerat et al, 2015), we note that this approach may result in some information loss.*

2. I also have an issue with the calculation of what you call "sharpness" and I think it comes back to a remark of reviewer #1 on using "sharpness versus precision". As indicated in Eq. 10, it is not the same as calculated in forecast verification studies, as presented in the reference books: Jolliffe, I.T., and D.B. Stephenson, 2012: Forecast Verification: A Practitioner's Guide in Atmospheric Science. 2nd Edition. Wiley and Sons Ltd, 274 pp. , and Wilks, D.S., 2011: Statistical Methods in the Atmospheric Sciences. 3rd Edition. Elsevier, 676 pp.

Sharpness is supposed to be an attribute that measures the degree of variability of the forecasts or concentration of the predictive distributions. In forecast verification, sharpness is a property of the forecast only. In Eq. 10, if I understand well, you calculate a type of coefficient of variation with the standard deviation of the time series of (median?) predictions and the mean of the observations. It could be understood, I guess, as a normalized "standard deviation", but I think it is misleading to call it "sharpness", and I suggest you to call lit otherwise to avoid misunderstanding from the forecast verification community.

*It is agreed that this difference should be highlighted to avoid misunderstanding. The measure of the degree of variability of the predictive distributions is considered to be captured by the standard deviation of the predictive distribution each time step, and then aggregated across time steps. We choose to normalise by the sum of the observed streamflow ($\sum_{t=1}^{N} \widetilde{Q_t}$), as this is consistent between model configurations and rainfall forcings. If this normalisation is not included there can be different baselines for the different model configurations or forcings, which can introduce biases in this metric (see McInerney et al., 2017). The name of the metric has been changed to highlight this normalisation step and avoid confusion, Sharpness_Norm. The text has been changed as follows:*

*__Sharpness__ refers to the width of the predictive distribution, and can otherwise be known as "resolution" or "precision". Typically, sharpness is a determined using the predicted values only. In this work a measure of sharpness (as the sum of the standard deviation of the predictions each time step), is normalised by the sum of the observed values, to enable a comparison of this metric across catchments with different magnitudes of flow. As such, sharpness is quantified using the following metric from McInerney et al. (2017):*

$$Sharpness_{Norm} = \sum_{t=1}^{N} sdev(\boldsymbol{Q}_t )/\sum_{t=1}^{N} \widetilde{Q_t} \qquad (10)$$

3) Minot issues are:

- Page 1, line 13 (abstract): consider changing to "… compared to a longer calibration period…"

*Corrected, "to" included.*

- Page 2, line 12: you explain in your reply to the reviewers that you prefer to use "forecast" instead of "forecasted". Maybe you should do the same here for the sake of consistency.

*Corrected, thank you for picking this up.*

- Page 2, line 15: I would consider changing to "… user need for monthly to seasonal streamflow forecasts…". In fact, the use of "seasonal" alone in the paper is appealing, but the study is dealing with "monthly" forecasts and not seasonal forecasts. I think you should not mix both and should keep to using "monthly" when referring to your study and application. For instance, I suggest to

consider this remark in these other parts of the paper: Page 4, line 12; Page 4, line 22; Page 17, line 8.

As pointed out, originally it was considered more streamlined to use "seasonal" in a more generic sense, referring to forecasts longer than a short term forecast (e.g. 7 days). However, it is agreed that this could be ambiguous, particularly as a season typically refers to multiple months. As such, the suggested change has been adopted, and "monthly" is used instead of "seasonal" when referring to the application.

- Page 2, line 20: consider changing to "… calibration; Mount et al., 2016; …)"

Corrected, extra brackets removed.

- Page 2, line 24: consider changing to "…(Wu et al; 2013); however this is not always the case (for example, Brigode et al., 2013)"

Corrected.

- Page 3, line 4: consider changing to "…etc.); see, for example, Westra et al., 2014)."

Corrected.

- Page 3, lines 10-11: consider changing to "…calibration period, which exposes….of a shorter calibration period, which exposes the model…"

Corrected.

- Page 3, line 17: consider changing to "…and Maranzo, 2004) and disaggregation approaches…"

Corrected.

- Page 4, line 4: "… forecasting applications in France."

Corrected.

- Page 4, line 17: consider changing to "... of this study are to:"

Corrected.

- Page 5, line 3: consider changing to "…(Figure 1), which conveys…"

Corrected.

- Page 8, line 5: consider replacing "big" to "important"

Corrected.

- Page 8, line 28: I would write "10-year calibration period" or "one-year warm-up period", etc.. Consider checking if that would be correct in English and change over all the uses in the paper.

The paper has been updated to include "-year" where appropriate, following a number.

- Page 11, line 13: with only one reference, I suggest deleting this part of the sentence "and are common in forecasting applications" and only mention "… monthly time scale (see another example in Lerat et al., 2015)".

Changed to:

*These choices of residual error models at the daily and monthly time scales contribute to the study objectives of reliable forecasts at the monthly time scale (see another example in Lerat et al., 2015).*

- Page 11, line 19: consider adding a comma: "For all scenarios, observed…"

Comma added.

- Equations on Pages 12-13: in general, I think the terms of the equations are not clearly presented. For instance, "Qt in bold" for predictions and "Qt~" for observations is only clearly stated in line 16 for Eq. 12, while it was already used in Eq. 10. Then in Eq. 14, we have a "Qt Teta": is it the same as "Qt in bold"?

Further explanation of the notation has been included in Section 3.5 and 3.6, in particular to explain $\boldsymbol{Q}_t$ clearly. $\boldsymbol{Q}_t$ represents the predictive distribution of the forecasts at time t (definition now added Page 11 line 5), and $Q_t^\theta$ the hydrological model predictions at month t, before application of the residual error model. As such, the different notation is considered necessary.

- Metrics: some metrics seem to sum/average over the time series before computation (Eq. 10 and 11), while other compute at each time step (or forecast monthly lead time) and then take the average (Eq. 12 and Eq. 14). Maybe it is worth untangling this issue when introducing the equations.

The metrics presented in Section 3.7 are generally computed each time step. The standard approach to calculating the volumetric bias has been presented for consistency, however this could be rearranged to take the difference between the expected and observed streamflow each time step, and then take the sum the differences. As outlined at comment 2, the sharpness is considered to be calculated at each time step (as the standard deviation) and then summed over each time step, with the sum over the observed values used to normalise across scenarios. As such, the general approach to compute each metric is to (i) calculate a metric value at each time step, (ii) sum/average over all time steps, and (iii) normalize to improve interpretation. The exception is the reliability metric, where we determine if the observations are considered to be samples from the predictive distribution, which can't be assessed separately at each time step.

- Page 13, line 16: "These changes…": what changes exactly are you referring to? Please, clarify.

This sentence ("These changes are considered in detail below") is considered unnecessary, as the changes are outlined in the following sentences. As such, it has been removed.

- Fig. 5: how do you explain the simulated peak flow at the end of 2002 that is not observed? Isn't there a problem with the observations? Could you say something about it in the text?

The following has been added to Section 5.4 (Future Work):

*These types of catchments are known to be challenging to model (McInerney et al., 2017; Ye et al., 1997). For example, the models predict a streamflow response in 2002 and 2005 in Figure 5 that did not occur in the observations, even when observed rainfall and state updating was used. Some of this difference may be due to errors in the input rainfall data, but this result highlights the difficulty in representing streamflow generation in low yielding, ephemeral catchments, such as those considered.*

- Fig. 8: for C1, at least, the parameter "split" seems to evolve around 0.9, therefore, finally, it is worth considering it as a free parameter to calibrate? Could you say something about it in the text?

It is considered further analysis would be required to evaluate the benefits of including the split parameter a free parameter to calibrate. For the longer calibration period length, split~0.95, and the

shorter calibration length the value changes over time from 0.95 to 0.8. Further work would be required to investigate if other parameters could have compensated from these changes from the default value of split=0.9, or the trends over time. No change is proposed.

- Discussion: please note that state updating was used for seasonal forecasting in "Crochemore, L., Ramos, M.-H., and Pappenberger, F., 2016: Bias correcting precipitation forecasts to improve the skill of seasonal streamflow forecasts, Hydrol. Earth Syst. Sci., 20, 3601-3618, doi:10.5194/hess-20-3601-2016". The authors also discuss about trade-offs between forecast attributes and it could be worth having a look at it.

Thank you for bringing this relevant study to our attention. The state updating approach used in this work is very similar to that used by Crochemore et al. (2016), and as such has been included in the manuscript here:

*The approach used for the state updating of GR4J is similar to the approach of Crochemore et al. (2016) and Demirel et al. (2013)*

The authors also discuss the trade-off between reliability and sharpness, and as such have also been cited here:

*All other metrics (sharpness, bias, CRPS and NSE) show improvements from state updating in catchment C2, suggesting potential trade-offs in performance, similar to that found by Crochemore et al. (2016) and McInerney et al. (2017).*

- Page 16, line 26: I am not sure about using the word "appreciably" here. I suggest deleting it or making more explicit what you mean here.

Removed.

- Page 17, line 8: "change to "These catchments were…"

Corrected.